# Population genomics uncovers loci for trait improvement in the indigenous African cereal tef (*Eragrostis tef*)
Maximillian R. W. Jones[1,11], Worku Kebede [2,3,11], Abel Teshome[1,4,11], Aiswarya Girija[5,6], Adanech Teshome[2], Dejene Girma[2], James K. M. Brown [1], Jesus Quiroz-Chavez[1], Chris S. Jones [7], Brande B. H. Wulff [8], Kebebew Assefa[2], Zerihun Tadele[9], Luis A. J. Mur [6], Solomon Chanyalew [2] ✉, Cristobal Uauy [1] ✉ & Oluwaseyi Shorinola [4,10] ✉

Tef (*Eragrostis tef*) is an indigenous African cereal that is gaining global attention as a gluten-free "superfood" with high protein, mineral, and fibre contents. However, tef yields are limited by lodging and by losses during harvest owing to its small grain size (150× lighter than wheat). Breeders must also consider a strong cultural preference for white-grained over brown-grained varieties. Tef is relatively understudied with limited "omics" resources. Here, we resequence 220 tef accessions from an Ethiopian diversity collection and also perform multi-locational phenotyping for 25 agronomic and grain traits. Grain metabolome profiling reveals differential accumulation of fatty acids and flavonoids between white and brown grains. *k*-mer and SNP-based genome-wide association uncover important marker-trait associations, including a significant 70 kb peak for panicle morphology containing the tef orthologue of rice *qSH1*—a transcription factor regulating inflorescence morphology in cereals. We also observe a previously unknown relationship between grain size, colour, and fatty acids. These traits are highly associated with retrotransposon insertions in homoeologues of *TRANSPARENT TESTA 2*, a known regulator of grain colour. Our study provides valuable resources for tef research and breeding, facilitating the development of improved cultivars with desirable agronomic and nutritional properties.

Tef (*Eragrostis tef* (*Zucc.*) Trotter) is a cereal crop that has been grown in the Horn of Africa for millennia. It is a self-pollinating allotetraploid grass that is valued by farmers as a 'fail-safe' cash crop, resilient to marginal soils, waterlogging, high temperatures, and drought. Tef is a staple crop in Ethiopia, Africa's second most populous country, where it is grown on 3 million hectares (27% of cereal acreage) by around 6.7 million households, with annual production exceeding 5.5 million tonnes[1]. The crop acts as both feed and food, with the straw a prized forage for cattle and the whole-grain flour used to produce a fermented flatbread known as injera, which serves as a staple food for the majority of the country[2].

Tef has also gained global attention as a 'superfood' thanks to its high protein, calcium, iron, and fibre contents, its low glycaemic index, and its lack of allergenic gluten[2]. Additionally, tef is rich in dietary antioxidants, including polyphenols and flavonoids, and essential polyunsaturated fatty acids like linoleic acid, which are not synthesised by the human body[3]. These nutritional features, combined with its climatic resilience, make tef an attractive crop for wider adoption. To date, government policies in Ethiopia have restricted export of tef germplasm and bulk grain to protect both its natural heritage and domestic consumption[4,5]. However, tef cultivation is expanding beyond Ethiopia, notably in the USA, Australia, South Africa,

[1]John Innes Centre, Norwich Research Park, Norwich, UK. [2]Ethiopian Institute of Agricultural Research (EIAR), Addis Ababa, Ethiopia. [3]Institute of Plant Sciences, Scuola Superiore Sant'Anna, Pisa, Italy. [4]International Livestock Research Institute (ILRI), Addis Ababa, Ethiopia. [5]Institute of Biological, Environmental & Rural Sciences (IBERS), Plas Gogerddan, Aberystwyth University, Ceredigion, UK. [6]Department of Life Sciences, Penglais Campus, Aberystwyth University, Aberystwyth, UK. [7]International Livestock Research Institute, Nairobi, Kenya. [8]Plant Science Program, Biological and Environmental Science and Engineering Division (BESE), King Abdullah University of Science and Technology (KAUST), Thuwal, Saudi Arabia. [9]University of Bern, Institute of Plant Sciences, Bern, Switzerland. [10]School of Biosciences, University of Birmingham, Birmingham, UK. [11]These authors contributed equally: Maximillian R. W. Jones, Worku Kebede, Abel Teshome. ✉e-mail: solchk2@gmail.com; cristobal.uauy@jic.ac.uk; o.shorinola@bham.ac.uk

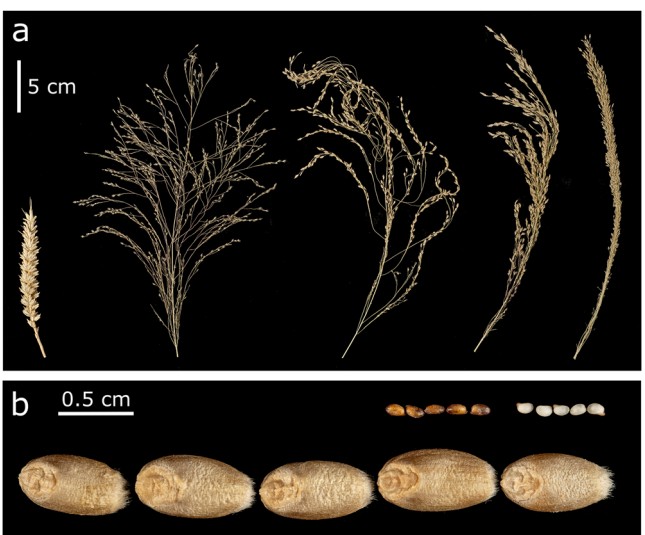

**Fig. 1 | Diversity of panicle morphology and grain colour in tef. a** Comparison of a bread wheat spike (cv. 'Paragon', far left) with tef accessions exemplifying four categories of panicle morphology (from left to right; very lax, lax, semi-compact, and compact). **b** Comparison of bread wheat grains (cv. Paragon, bottom) with grains from brown and white-grained tef varieties.

and the Mediterranean regions[6,7]. In these areas, tef is also used as a multi-harvest forage crop for producing premium-quality hay and silage[8].

Tef is considered an underutilised crop because it has, so far, not benefited greatly from modern genomics-based approaches to breeding and research. However, as in other underutilised crops such as grass pea, yam, and lablab, this *status quo* is beginning to shift[9,10], with the generation of a high-quality reference genome[11] following on from a draft sequence[12]. Notable progress has been made in breeding improved varieties of tef[2,13], although advances have not been on the same scale as for major cereals like wheat or rice. Lodging under high nitrogenous fertiliser regimes is a major limiting factor but has so far been difficult to address through classical semi-dwarfing approaches, at least partially because of the value of tef straw as animal feed. Addressing lodging through improved root traits is also being explored[14,15]. Panicle (inflorescence) morphology has been reported as a determinant of lodging tolerance in tef. The species exhibits dramatic panicle diversity[16], from open, highly lax panicles similar to wild *Eragrostis* species, to short-branched, compact panicles more akin to the spikes of Triticeae species (Fig. 1a). Using a combination of controlled-environment phenotyping, mechanical testing, and crop modelling, Blosch et al. showed that tef varieties with compact panicles tended to be more resistant to lodging, suggesting an ideotype approach could be used to address this issue[17].

Another consideration for breeders is grain size. Tef produces the smallest grains of any cultivated cereal, ranging from 1.0 to 1.7 mm in length, with a typical thousand grain weight (TGW) of 0.2–0.4 g, roughly 150-fold lower than that of wheat[2,18,19] (Fig. 1b). Indeed, the name tef is thought to derive from the Amharic word "teffa" meaning "lost"[20]; likely an allusion to the high levels of harvest and post-harvest losses (16–30%) experienced by tef farmers[6,21]. Breeding for larger grains could alleviate these losses and boost realised yields by improving the separation of grain and chaff during winnowing. Tef's small grain size also makes it difficult to evenly broadcast the recommended >10 kg/ha during sowing. Farmers instead use high sowing rates (up to 30 kg/ha) that ultimately produce overcrowded fields prone to lodging[22]. Lastly, breeders must also address a strong cultural preference for white-grained over brown-grained varieties, which translates into a higher market price for the former[23].

Here, we aim to use a population genomics approach to study the diversity and genetic architecture of agronomic, grain morphology, and

grain metabolite traits in a representative Ethiopian tef collection. We therefore conducted short-read resequencing of 220 tef accessions from the Ethiopian Institute of Agricultural Research (EIAR) tef diversity collection. We characterised redundancy in this collection and produced a compact SNP panel that uniquely identifies the studied accessions. We combined this sequencing data with extensive in-field phenotyping across three trial locations, including precise grain morphology measurements and grain metabolome profiling (Fig. 2). This led to the identification of important marker-trait associations for panicle morphology, grain size, grain colour, and multiple grain metabolites. Our analyses establish a previously unknown link between grain size and grain colour, including the co-association of these traits with multiple genomic loci. However, we also identify regions that decouple these traits, offering potential breeding opportunities. Our work delivers a set of genomic and phenotypic resources for a diverse panel of tef accessions and lays the groundwork for future studies to define causal genes and variants underlying loci of agronomic relevance.

## Results

### SNP and *k*-mer-based methods identify redundancy in the EIAR core collection

Of the 225 accessions in the EIAR core collection, we sequenced (Illumina paired-end 150 bp) the genome of 220 accessions to an average depth of 8.85 Gbp (SD = 0.77 Gbp), equivalent to 14.2-fold the estimated genome size (622 Mb) of the reference genome cultivar 'Dabbi'[11]. The reads were mapped against the reference genome and variants were called. A quality-filtered and linkage-pruned set of 41,289 SNPs was prepared and used to investigate linkage disequilibrium (LD) decay and population structure. The genome-wide LD decay distance in the EIAR core collection was 46.3 kb (Supplementary Fig. 1). ADMIXTURE analysis revealed no clear population structure for two to 20 subpopulations[24]. Principal component analysis also did not suggest any distinct lineages, although there was a partial separation of brown and white-grained accessions by PC1 (Supplementary Fig. 1). This result was not unexpected, as a lack of strong population structure has previously been reported for other tef panels[25].

An SNP-based phylogram was computed using IQ-TREE 2. This revealed many groups of highly related accessions, with internal branch lengths close to zero nucleotide substitutions per site (Fig. 3a)[26]. A list of redundancy groups was defined such that the total branch length (phylogenetic distance) between any pair of accessions in a group was <0.005 substitutions per base pair. This resulted in 31 redundancy groups containing 2–19 accessions per group (Supplementary Table 1). Two accessions were excluded from placement in redundancy groups because they had high apparent heterozygosities (22.7% and 24.9% heterozygous sites, versus an average of 1.5% (standard deviation (SD) = 0.7%) for the other accessions) and likely represent seed mixtures rather than pure accessions.

To validate the redundancy groups defined above, a comprehensive *k*-mer (k = 51) presence/absence matrix was generated from the sequencing reads of the 220 sequenced accessions[27]. We tested whether pairs of accessions from the previously defined groups tended to have more *k*-mer states in common than non-group pairs (Fig. 3b, c, Supplementary Data 1). We observed that all 264 intra-group pairs had a *k*-mer state identity rate above 96.0%, whereas 23,825 out of the 23,826 other pairs had a *k*-mer state identity rate below this threshold, with a mean and median of 85.4% (SD = 0.02%). The single pair exceeding this threshold (DZ-01-91 and DZ-01-101), still had a very low phylogenetic distance (0.007) compared to the overall distribution, so were added as an additional redundancy group. The broad agreement between these two relatedness metrics suggested that the redundancy groups would be better treated as single accession pools rather than distinct entities. This reduced our effective number of accessions from 220 to 150.

One accession, DZ-01-1167, produced notably low shared *k*-mer state rates when paired against all other accessions (Fig. 3b, below the dashed line). DZ-01-1167 contains 294 million distinct *k*-mers versus an average of

**Fig. 2 | Resequencing, phenotyping, and GWAS of the EIAR core tef collection.** A representative panel of Ethiopian tef accessions was resequenced and phenotyped for agronomic, grain size, and metabolomic traits. Statistical modelling was used to correct for location and within-site spatial effects. The software and numbers of accessions used for each step are indicated above each box. Vector images of the tef plant and sequencer were created by Wanda Pelin Canila/Shutterstock.com and Jaitham/Shutterstock.com, respectively.

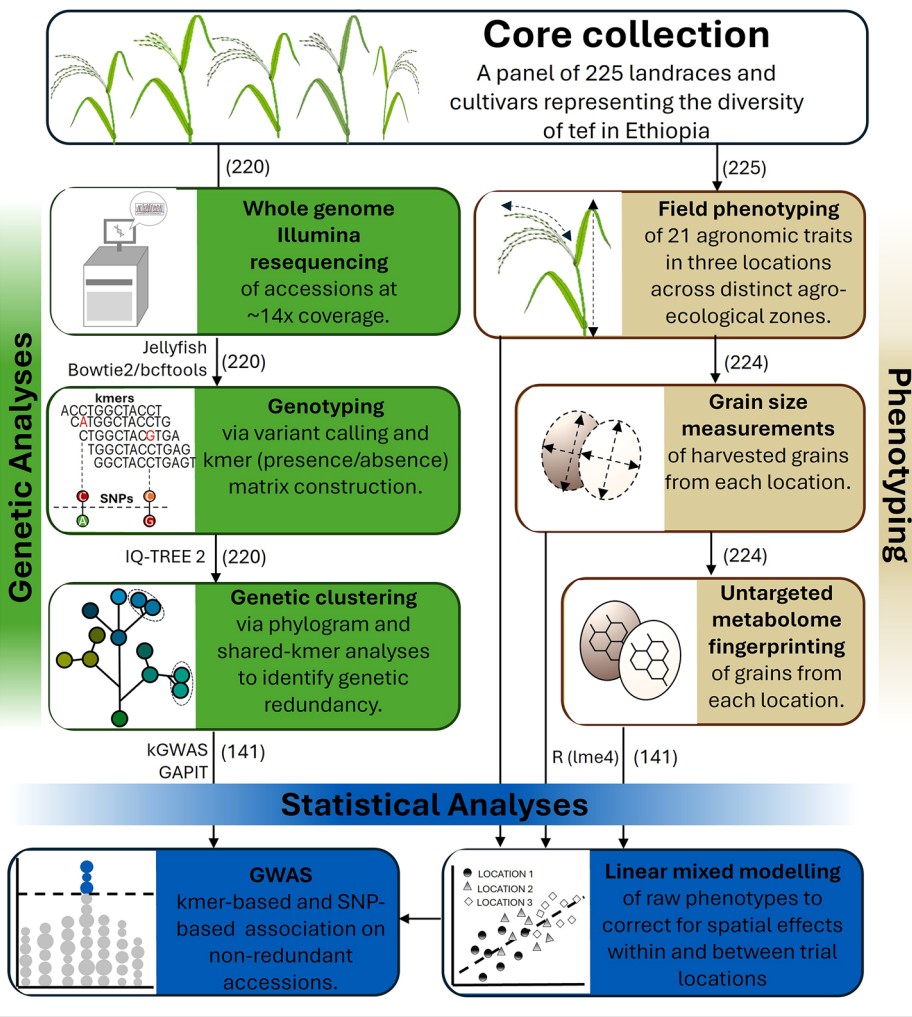

160 million (SD = 3.5 million) for all other accessions (Supplementary Fig. 3), suggesting it contains many unique *k*-mers. This new diversity could derive from genetically distant tef accessions not otherwise captured in the panel or from interspecific introgression(s). Its source and utility could be further explored in future studies. This finding highlights the benefits of *k*-mer-based approaches, as this introgression is not apparent when comparing SNP-based phylogenetic distances.

Given the high levels of redundancy observed in the EIAR core collection, we selected a minimal panel of SNPs capable of distinguishing all 150 accession groups and singlets. This would allow other accessions belonging to the redundancy groups to be identified amongst the wider EIAR collection, as well as potentially facilitating some reconciliation with other tef collections. We identified a panel of 14 biallelic SNPs that could distinguish all 150 accession groups or singlets. To account for potential marker failures, we selected an additional 14 SNPs, making a total of 28. For each of these SNPs, all accession groups and singlets were homozygous for one allele or had missing data (Supplementary Data 2, Supplementary Note 1). All chromosomes were represented by at least one SNP except 5A, 7B, 8B, and 10A.

### Grain colour strongly correlates with plant height and grain morphometric traits

To capture phenotypic variation in the EIAR core collection, we phenotyped 17 phenological and morphological traits at three field sites representing distinct agro-ecological zones (Supplementary Figs. 4, 5). Using high-resolution grain imaging, we also captured variation in eight grain size parameters relating to grain width, length and area. There was a significant effect (ANOVA *p*-value < 0.04) of location on all traits

except for grain width and length. Trait coefficients of variation at each location ranged from 0.02 to 0.45, with phenology traits showing the least variation.

To account for spatial variability within and between experimental locations, we used linear mixed modelling to generate genotypic best linear unbiased predictors (BLUPs) and broad-sense heritabilities ($H^2$) for each trait. The heritability for agronomic and grain morphometric traits ranged from 0.14 to 0.97 (Supplementary Table 2). As expected, qualitative traits like panicle form and grain colour showed the highest heritability, 0.92 and 0.97, respectively. The heritability for grain morphometric traits, including grain length, width and area, was also high: 0.87, 0.90, and 0.89, respectively. Correlation analysis of trait BLUPs revealed expected associations between components of grain size (area, width, length) and weight, as well as between components of plant height (panicle length, plant height) (Fig. 4a).

Intriguingly, we also identified strong correlations between grain colour and both grain size and plant height. Plants of white-grained varieties were significantly taller than those of brown-grained varieties (Student's *t*-test, $p < 1 \times 10^{-5}$, Fig. 4b). This has positive implications for straw yields, but may increase lodging susceptibility. Brown-grained accessions are traditionally cultivated on more marginal soils, such as poorly drained vertisols, and, perhaps as a result, have come to be associated with smaller grains. However, our results indicated that brown-grained varieties tended to produce larger seeds than white-grained varieties when grown in common environments (Student's *t*-test, $p < 1 \times 10^{-15}$, Fig. 4c). Despite this, there was no difference in TGW between white and brown-grained varieties (Student's *t*-test, $p = 0.22$), suggesting that, on average, white-grained varieties produce grains with higher densities.

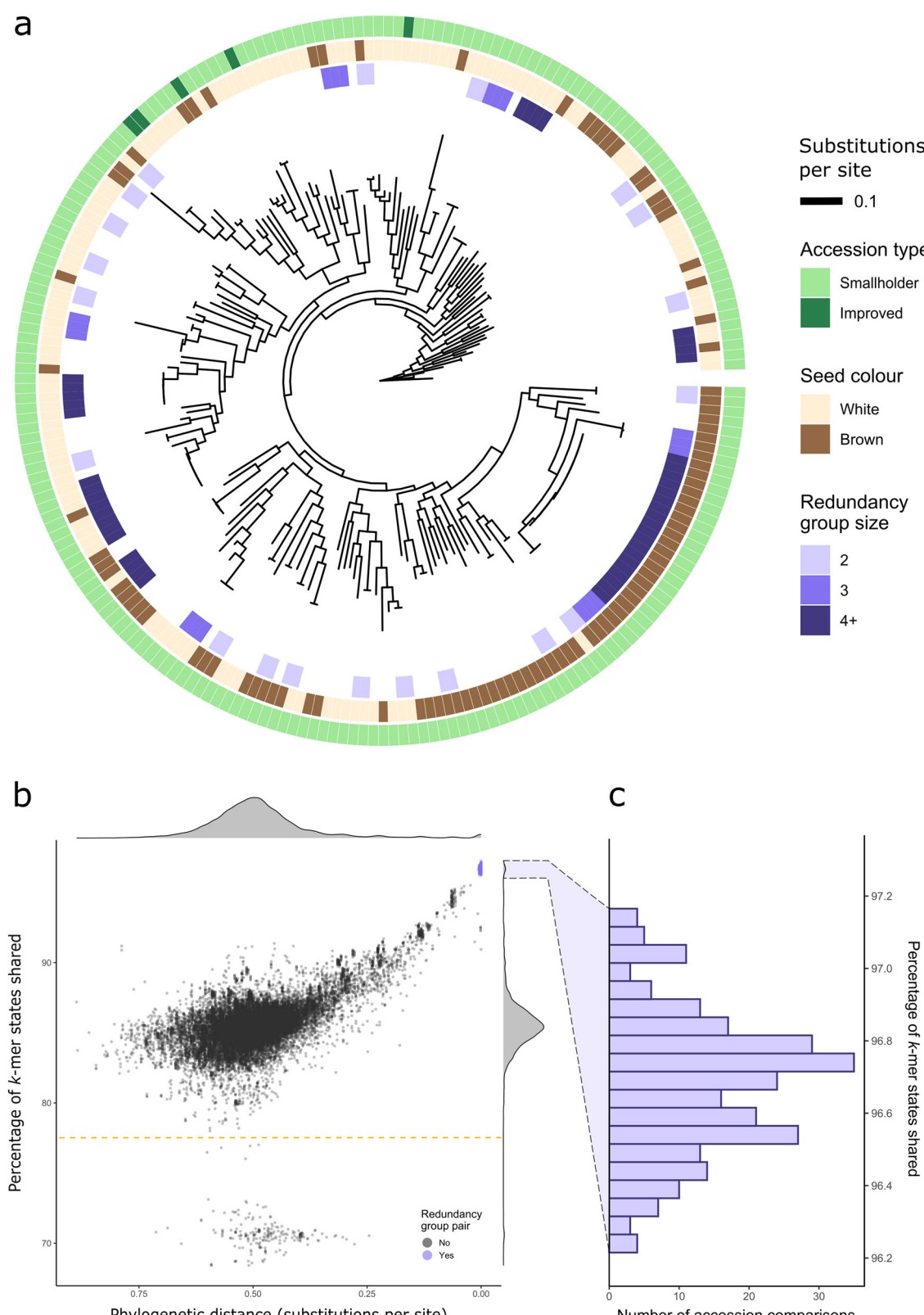

**Fig. 3 | Phylogenetic analyses identify redundancy in the EIAR core collection.**
**a** Phylogram of 220 tef accessions, arbitrarily rooted against the accession "Ada-T58". White and brown-grained varieties are well-distributed across the phylogeny. A total of 32 redundancy groups, ranging in size from 2 to 19 accessions, were identified on the basis of small phylogenetic distances between pairs of accessions.
**b** Phylogenetic distance plotted against the percentage of $k$-mer states shared for all 24,090 pairwise comparisons between accessions. There is a strong correlation between these two relatedness metrics. Accession pairs from within the previously defined redundancy groups (purple) cluster together at uniquely high shared $k$-mer state rates, depicted in detail in (**c**). The points with particularly low percentages of shared $k$-mer states (<78%, dashed line) represent the full set of comparisons of "DZ-01-1167" with other accessions.

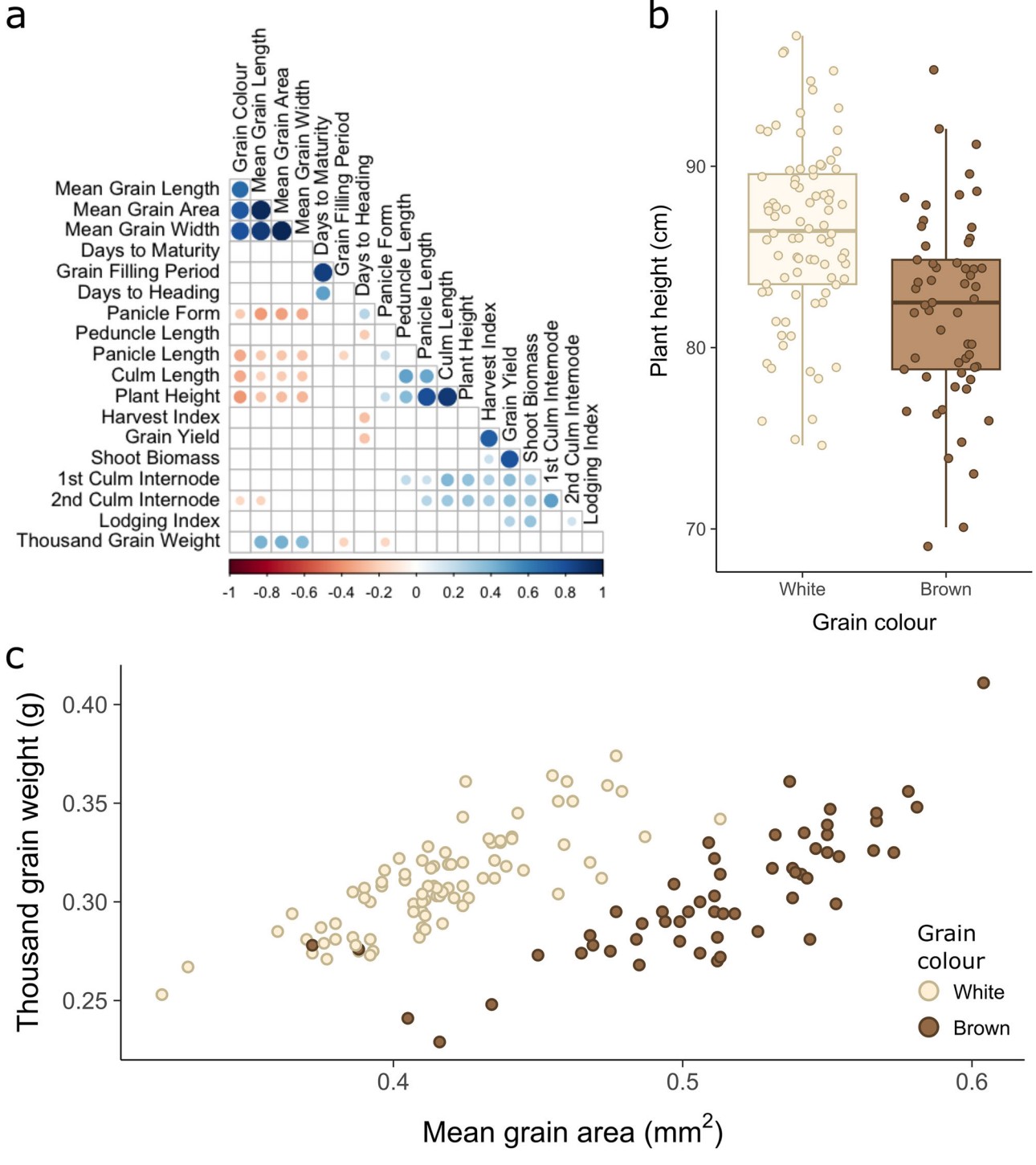

**Fig. 4 | Best linear unbiased predictors (BLUPs) reveal correlations between key agronomic traits. a–c** Analysis of BLUPs for *n* = 141 accessions and redundancy groups. **a** Correlation tests were conducted between the BLUPs for 20 traits of interest. Significant correlations (*p* < 0.05) are indicated by circles whose size and colour represent the magnitude and direction of correlation. **b** Boxplot of plant height BLUPs for white-grained (*n* = 84) and brown-grained (*n* = 57) varieties, with individual data points overlaid. White-grained accessions tended to produce taller plants. The centre line represents the median; the lower and upper hinges correspond to the 25th and 75th percentiles, and the whisker extends to 1.5 * Interquantile range (IQR). **c** Scatterplot of grain area BLUPs against thousand grain weight (TGW) BLUPs. A distinctly bimodal distribution is strongly explained by grain colour, with white-grained varieties tending to produce smaller grains. Despite this, their grains are of approximately the same mass as brown-grained varieties, suggesting higher grain densities.

Lodging index was, as expected, positively correlated with aboveground biomass, highlighting the trade-off faced by tef breeders between lodging rates and straw yields. However, we did not observe the previously reported correlation between lodging index and panicle morphology. This could be due to the relatively few lines (five) with the highest level of panicle compactness or the lower robustness of our lodging index BLUPs, given that this trait was only phenotyped at two of the three field sites.

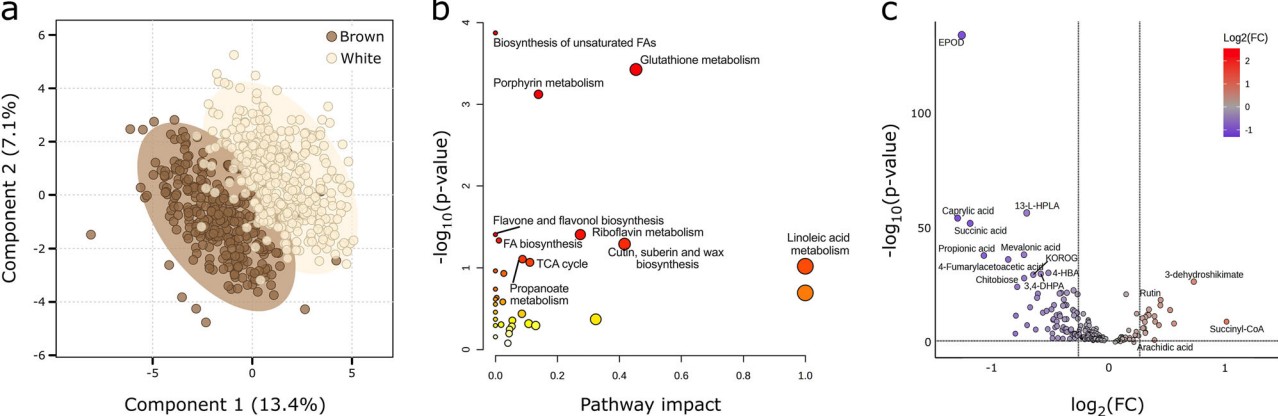

**Fig. 5 | Brown and white-grained tef accessions display differential metabolite accumulation. a** Partial least squares discriminant analysis (PLS-DA) of metabolites in grain samples of brown and white-grained accessions. Ellipses represent 95% confidence intervals around each group. **b** Differentially accumulated metabolites show enrichment for several metabolic pathways, notably fatty acid and flavone metabolism. Point size scales with pathway impact and colour intensity scales with significance of pathway enrichment. **c** Volcano plot for the 183 identifiable differentially accumulated metabolites. Fold-change (FC) was calculated as mean value in brown-grained varieties divided by that in white-grained varieties. Plotted FC thresholds are $\log_2(0.83)$ and $\log_2(1.2)$ and plotted FDR threshold is $\log_{10}(0.05)$.

## High-resolution metabolite fingerprinting shows differential metabolite accumulation in brown and white tef grains

The cultural preference for white-grained varieties in Ethiopia and Eritrea, as well as the growing international interest in tef's nutritional properties, motivated us to also explore variation in tef's grain metabolomes. We performed untargeted metabolite profiling using Flow Infusion Electrospray High-Resolution Mass Spectrometry (FIE-HRMS) analysis on grain samples from each plot of the three trial locations. A total of 1643 positively ionised mass-to-charge ratio ($m/z$) features and 1470 negatively ionised features were captured, of which 209 and 723, respectively, were differentially accumulated in brown and white-grained varieties (Student's $t$-test, FDR < 0.05).

From these differential $m/z$ features, 183 could be tentatively identified using a rice (*Oryza sativa* ssp. *japonica*) reference metabolome library available in the KEGG public database[28] (Supplementary Data 3). These differentially accumulated metabolites (DAMs) produced a clear separation of white and brown-grain samples when assessed by partial least squares discriminant analysis (PLS-DA) (Fig. 5a), but did not show differential accumulation between locations, suggesting little effect of locations on these metabolites (Supplementary Fig. 6). The DAMs were enriched for various processes, including (unsaturated) fatty acid biosynthesis, linoleic acid metabolism, glutathione metabolism, porphyrin metabolism, flavone, and flavonol biosynthesis, and riboflavin metabolism (Fig. 5b).

In agreement with other studies[3], our results show that brown-grained varieties tend to have higher proportions of essential polyunsaturated omega-6 and omega-3 fatty acids (e.g., linoleic acid and alpha-linolenic acid, respectively) while white-grained varieties have higher levels of saturated fatty acids (e.g., caprylic acid and 9,10-epoxyoctadecanoic acid (EPOD)). Omega fatty acids have been associated with lowering cardiovascular disease, cancer, and autoimmune diseases[29]. However, the high levels of unsaturated fatty acids in brown-grained tef may also contribute to its increased proneness to rancidity and therefore its lower consumer appeal[30] (Fig. 5c, Supplementary Fig. 7a).

We also found differential accumulation of flavonoids between white and brown-grained varieties. These compounds can affect flavour and colour and act as antioxidants[31]. We found increased levels of the flavonoids rutin and 3-dehydroshikimate in brown-grained varieties. Meanwhile, in white-grained varieties, we observed elevated levels of apigenin and kaempferol 3-O-rhamnoside-7-O-glucoside (KOROG). Flavonols such as the latter have been known to contribute to white pigmentation[32] (Fig. 5c, Supplementary Fig. 7b).

## *k*-mer-based GWAS identifies regions associated with panicle and grain morphologies

To identify genomic regions associated with the agronomic and grain morphology traits, we conducted a *k*-mer-based genome-wide association study (kGWAS) using the previously calculated BLUPs and a new *k*-mer matrix to account for the reduced number of non-redundant accessions. Of the agronomic traits tested, we detected significant marker-trait associations (MTAs) for panicle morphology, grain morphology, and grain colour (Supplementary Table 3). We also carried out an SNP-based GWAS and identified four significant MTAs for panicle morphology, grain morphology, and lodging index (Supplementary Table 4).

The control of panicle morphology appeared to be relatively simple, with a single highly associated 70-kb region on chromosome 3B (Fig. 6b). The underlying reference *k*-mers negatively correlated with panicle morphology (scored as 1 to 4 for very lax to compact). This matched our expectations as the reference cultivar Dabbi produces very lax panicles. The significant region contains 13 gene models, including the tef orthologue (Et_3B_031395) of the rice gene *qSH1/RIL1* (*QTL for Seed Shattering on chromosome 1/RI-LIKE1*; Os01g0848400). *qSH1* is a *BEL1*-like homeobox transcription factor linked with seed shattering and inflorescence architecture[33,34]. *qSH1* orthologues are also expressed in the inflorescence meristems of maize and wheat, suggesting a conserved role in inflorescence development across the grass family [35,36].

There was a set of complex co-localised associations for grain morphology (Fig. 6a). Most strikingly, a highly significant region (peak 7) supported by tens of thousands of *k*-mers was detected on chromosome 4B for grain colour and grain width (Fig. 7a, b). This region was, as expected, also associated with grain area, but was not significant for grain length (Fig. 7b, Supplementary Fig. 8a, b). There was also a region on chromosome 4A (peak 3) significantly associated with grain colour and width (Fig. 7a, b). In addition, we found smaller regions on chromosome 3B (peak 2) and 4A (peak 6) co-associated with grain colour and grain width (Figs. 6a, 7a, b). The *k*-mers in each of the above regions were correlated with brown grain colour and increased grain size parameters. This supports the correlation between grain colour and grain size observed in the plotted BLUPs (Fig. 4c).

In contrast to the regions discussed above, there were also cases where grain colour and grain size were decoupled. A third positive grain width peak on chromosome 4A (Fig. 7b peak 5) is well-separated from the upstream and downstream grain colour peaks, by 8460 kb and 1580 kb, respectively. On chromosome 4B there is a region negatively associated with grain width and area (Fig. 7b peak

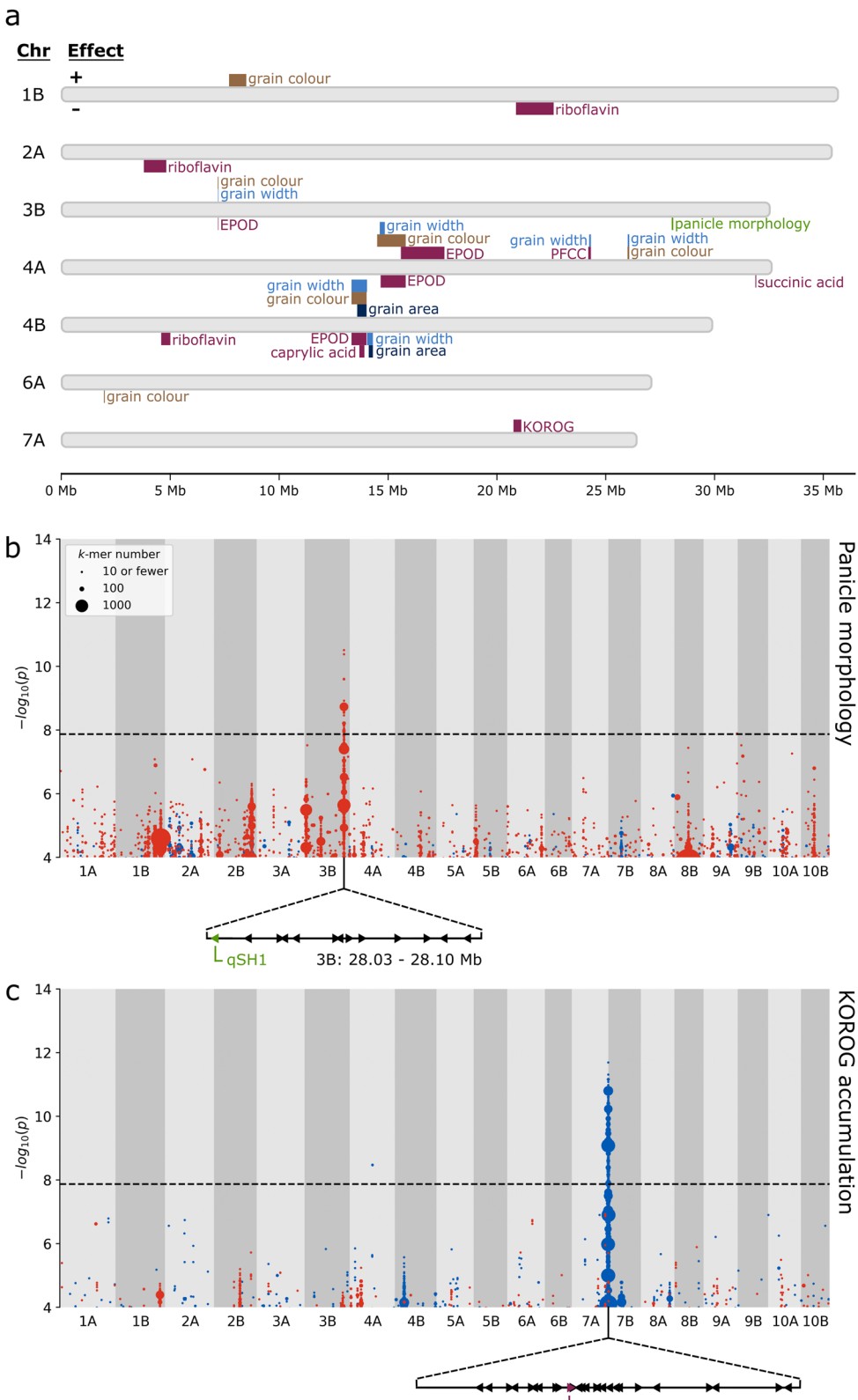

**Fig. 6 | *k*-mer-based GWAS identifies multiple marker-trait associations, including regions associated with panicle morphology and grain KOROG. a** Plot summarising all trait-associated regions identified by *k*-mer-based GWAS. Regions positively associated with traits are plotted above their respective chromosomes, while negatively associated regions are plotted below. For grain colour, positive and negative associations indicate brown and white, respectively. **b** A region significantly associated with panicle morphology was detected on chromosome 3B. The arrangement of the 13 genes within this region is displayed below the plot. The candidate gene *qSH1* is highlighted. **c** A region significantly associated with Kaempferol 3-O-rhamnoside-7-O-glucoside (KOROG) was detected on chromosome 7A. The arrangement of the 26 genes within this region is displayed below the plot. The candidate gene *CYP93G1* is highlighted. In (**b** and **c**), *k*-mers are grouped according to their association level and genomic coordinates (10 kb bins) and coloured according to the direction of association; red for panicle laxness or low KOROG, blue for panicle compactness or high KOROG. Point size is proportional to the number of *k*-mers rounded upwards to the nearest 10.

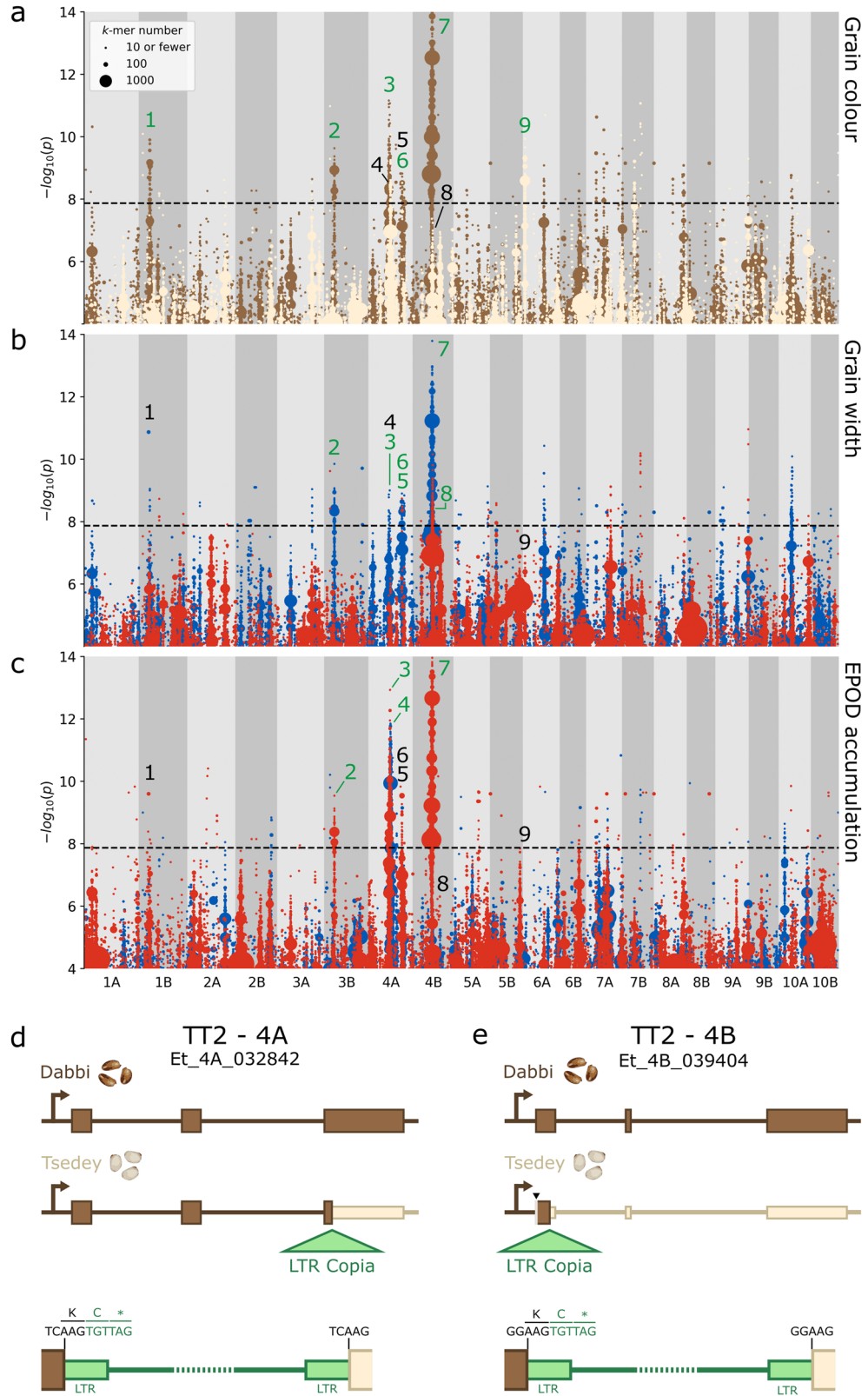

8) that is separated by just 10 kb from peak 7, a major peak for brown grain colour and higher grain width and area. A marginally insignificant peak for grain width and area exists on chromosome 10A (Fig. 7b, Supplementary Fig. 8). Lastly, there are two grain colour peaks on chromosomes with no significant grain size peaks (Fig. 7a).

This includes a 790 kb peak associated with brown grains on chromosome 1B (114 genes) and a 40 kb peak associated with white grains on chromosome 6A (6 genes). These two regions offer the strong possibility of breeding for grain colour independently of grain size.

**Fig. 7 | Co-association of grain colour, width, and EPOD concentration with multiple regions.** Plots of *k*-mers associated with **a** grain colour, **b** grain width, and **c** grain EPOD concentration. *k*-mers are grouped according to their association level and genomic coordinates (10 kb bins) and coloured according to the direction of association. In (**a**), brown denotes association with brown grain colour and white with white grain colour. In (**b** and **c**), red denotes association with lower trait values, and blue with higher trait values. Point size is proportional to the number of *k*-mers rounded upwards to the nearest 10. Nine regions are labelled with black and green numbers, denoting whether the region is significant or not significant for the plotted trait, respectively. Diagrams of LTR Copia insertions into *TT2* homoeologs on **d** chromosome 4A, **e** chromosome 4B. Top: structure of *TT2* in Dabbi (brown-grained). Centre: structure of *TT2* in Tsedey (white-grained). Bottom: detail of LTR Copia insertions. Narrower exons indicate presumed protein truncations. Black DNA bases denote 5 bp target-site duplications. Green DNA bases denote the start of the retrotransposon insertions. Single-letter amino acid codes show the introduction of premature stop codons (*). The first 22 bp of the *TT2* open reading frame on 4B is not assembled in the Tsedey genome (greyed out, black arrowhead). Gene annotations derive from ref. 11. and do not include 5' and 3' untranslated regions.

## Associated regions for grain metabolites and grain morphology co-localise

We hypothesised that genomic loci associated with grain colour might also be associated with the differentially accumulated metabolites. To facilitate GWAS analysis, we calculated BLUPs and broad-sense heritabilities for 21 grain metabolites with high fold-change differences and/or which are known to be involved in pigmentation or important for human nutrition (Supplementary Data 3). Heritability values were generally high, with 18 of the 21 metabolites having $H^2 > 0.50$ (Supplementary Table 5). kGWAS revealed significant regions for riboflavin, EPOD, primary fluorescent chlorophyll catabolite (PFCC), succinic acid, caprylic acid, and KOROG (Fig. 6a, Supplementary Table 3). SNP-based GWAS also identified nine MTA for four metabolite traits, five of which overlap with the significant region for kGWAS (Supplementary Table 4).

As we hypothesised, kGWAS associations for some metabolites, including EPOD, caprylic acid and PFCC, co-localise with regions associated with grain colour and/or size. Of these, EPOD showed the most consistent and significant overlap with grain colour and grain width, on chromosomes 3B (peak 2), 4A (peak 3) and 4B (peak 7, also overlapping a grain area peak) (Fig. 7c). In these co-localised regions, EPOD was negatively associated with grain colour (i.e brown-grained varieties had lower EPOD content). There was also a significant peak for EPOD on chromosome 4A (peak 4) that partially overlapped, and was positively correlated with, a grain colour peak (peak 3). The 4A peaks for EPOD (peaks 3 and 4) were also identified in our SNP-based GWAS. Caprylic acid was associated with a single locus that overlapped with peak 7 for EPOD, grain colour, and grain width. Similarly, PFCC was associated with a single region in both *k*-mer and SNP-based GWAS that also overlapped with grain width.

We also found associations for other metabolites, including KOROG, succinic acid, and riboflavin, that did not overlap with grain colour and/or size. KOROG was associated in both *k*-mer and SNP-based GWAS with a single prominent 360 kb region on chromosome 7A (Fig. 6c) containing 26 gene models. This included Et_7A_050580, which encodes a cytochrome P450 (*CYP93G1*) with flavanone 2-hydroxylase (F2H) activity and has been previously shown to be involved in the biosynthesis of flavonol glycosides like KOROG[37]. Succinic acid was also associated with a single region, in this case spanning 40 kb on chromosome 4A and containing eleven genes. Control of riboflavin accumulation appears more complex, with three regions on chromosomes 1B (1730 kb, 174 genes), 2A (1030 kb, 126 genes), and 4B (420 kb, 75 genes) associated with riboflavin accumulation but that are not associated with other traits.

## *TRANSPARENT TESTA 2* is a candidate for grain colour variation

Given that the most prominent associations for grain colour, size and metabolites content cluster at peak 3 (chr 4A: 14.48–15.79 Mb) and peak 7 (chr 4B: 13.34–14.05 Mb) (Figs. 6a, 7a, b, Supplementary Table 3), we examined the gene content in these peaks to identify potential candidate genes. Interestingly, these peaks displayed partial homology; genes in the proximal end of peak 3 are homoeologous to genes in the distal end of peak 7 (Supplementary Fig. 9). Homoeologous gene pairs in these regions include Et_4A_032844/Et_4B_037039 and Et_4A_032842/Et_4B_039404, whose orthologues have been previously shown to regulate grain colour, size and fatty acid content. The former are orthologues of *NUCLEAR FACTOR YA3* (*NF-YA3*), which regulates seed oil content and seed size across diverse angiosperms, including *Arabidopsis* and oil palm[38]. The latter are orthologues of *TRANSPARENT TESTA 2* (*TT2*), a MYB transcription factor that is associated with proanthocyanidin accumulation in the seeds of various Brassicaceae species[39,40].

We compared the gene sequences of the *TT2* orthologues from the Dabbi reference genome (a brown-grained variety) to those from the published draft assembly of the white-grained variety 'Tsedey'[12]. In the Tsedey assembly, we identified striking insertions of long-terminal repeat (LTR) Copia superfamily retrotransposons (RTs) in the third and first exons of the A and B *TT2* homoeologues, respectively (Fig. 7d, e, Supplementary Data 4 and 5). The A-subgenome RT introduces an in-frame cysteine and then a premature stop codon, truncating most of the final exon (162 codons). The B-subgenome RT is inserted in the first exon, truncating most of the protein. The positions of the two elements in different exons suggest independent insertions occurring after subgenome divergence. In both RTs, the 5 bp target-site duplications and 106 bp LTRs remain undegraded, suggesting a relatively recent insertion[41]. This is consistent with the large number of recently active LTR RT families previously identified in tef. We did not find any protein-truncating mutation between Dabbi and Tsedey in the A and B homoeologues of *NF-YA3*; the B homoeologue contains one non-deleterious missense mutation, while the A homoeologue contains no missense mutation.

## Discussion

Developing genomic resources for underutilised crops is crucial for accelerating their improvement, adoption, and utilisation, and will in turn boost the resilience of the interconnected global food system[10,42]. Our extensive phenotyping and whole genome resequencing of a diverse tef collection represents a valuable resource for germplasm characterisation and trait mapping in this locally vital and globally emerging crop.

While grown in Ethiopia and Eritrea for thousands of years, systematic collection and breeding of tef did not begin until the 1950s, with sampling of varieties directly from farmers' fields. Since then, numerous germplasm collections have been established, containing over 7000 accessions. These are primarily maintained in Ethiopia, but smaller collections exist elsewhere[18,43]. Correspondence of varieties between these collections is undocumented, preventing cross-utilisation of phenotyping and sequencing data. Genomics approaches have been invaluable for resolving such issues and for identifying redundancy within collections[44,45]. Our work reveals redundancies in the EIAR tef collection, which are likely due to repeated sampling of farmer-traded germplasm across modest geographical ranges. The compact SNP panel we have developed can be used to identify further redundancy within the EIAR collection. The resequencing data presented here can also be combined with existing mid-density genotyping data from other tef collections to assess redundancies, differences, and complementarity between the different tef collections globally[25,46]. This will shed further light on tef's breeding history, facilitate germplasm exchange, and inform the selection of accessions for a tef pan-genome to optimally capture tef diversity.

We identified a strong candidate gene for panicle morphology, Et_3B_031395, which is orthologous to the rice *qSH1* gene. qSH1 is a BEL1-like homeodomain protein that underlies variation in seed shattering in rice[33] and is regulated by *SUPERNUMERARY BRACT*[47], whose direct orthologue in wheat is the major inflorescence morphology gene *Q*, which controls both seed shattering and inflorescence compactness[48]. The *Arabidopsis* orthologue of *qSH1*, *REPLUMLESS/PENNY*[49], also known as

*PENNYWISE*[50] or *BELLRINGER*[51], is important for fruit development and dehiscence. *qSH1* has also been directly connected with inflorescence architecture, with *qsh1 ri* (*verticillate rachis*) double mutants displaying abnormal timing and arrangement of primary branch meristems[33,34]. *qSH1* also strongly influences bract suppression in the inflorescence[33,34]. In addition, paralogs and orthologues of *qSH1* have been shown to control inflorescence patterning in maize and *Arabidopsis*[50–52]. Given the established roles of its orthologues in diverse plant species and its localisation within a narrow candidate region, Et_3B_031395 emerges as a promising candidate for the regulation of panicle morphology in tef. Nonetheless, functional validation will be necessary to confirm its role.

Most farmers in Ethiopia rely on manual broadcasting for sowing on small plots (0.25 to 1 hectare), as access to mechanisation is limited and such equipment is typically not optimised for the tiny seeds of tef[53,54]. This practice leads to inefficient seed use as farmers typically sow at higher rates than recommended to ensure good field coverage (15–25 kg/ha, instead of 5 kg/ha)[55]. These high seeding rates also produce over-crowded fields of weak-stemmed plants more prone to lodging[22]. Larger grains would facilitate mechanised handling and ensure seedlings have sufficient nutrients for establishment from greater soil depths. Together, this would promote row-based drilling of tef, alleviating the above issues and supporting the ongoing transformation of tef cultivation practices. Indeed, agronomists have already experimented with pelleting tef seeds with inert material to enable mechanised sowing, highlighting the promise of this approach[56]. Lastly, increasing grain size could help reduce the high grain loss rates experienced by smallholders during traditional threshing and winnowing processes[21].

Our kGWAS results offer potential breeding targets for increasing grain size. We identified six genomic regions significantly associated with grain width, two of which were also associated with grain area. However, we also identified a previously unknown link between grain size and grain colour, which could complicate this process. Brown-grained varieties tended to produce larger grains (0.51 mm$^2$) than white-grained varieties (0.42 mm$^2$), and this was reflected in the kGWAS; four of the regions positively associated with grain width were also associated with brown grain colour. This co-localisation presents an issue because there is a strong cultural preference in Ethiopia for white tef flour, and this translates into a market incentive for farmers to grow white-grained varieties. Introgression of grain size alleles into elite white-grained varieties at the cost of increased pigmentation would therefore not be favourable.

However, not all grain size and grain colour loci were co-localised. We identified one grain width locus on chromosome 4A that is very distant from the two grain colour regions on this chromosome, plus two grain colour loci on chromosomes which do not harbour grain size loci (1B and 6A). We also observed a region on chromosome 7A strongly associated with the metabolite KOROG. While our analysis did not find this region to be co-associated with grain colour, flavonols such as KOROG have previously been associated with white pigmentation[32]. These regions offer opportunities to combine favourable grain size and colour alleles through introgression, though it is yet to be seen if the introduction of 'white grain' alleles into a brown-grained background would yield a dominant effect. It is more likely that positive breeding outcomes could be achieved by stacking multiple additive grain size loci. Uncovering further such loci should be a priority for future GWAS studies in tef. It is also important to note that the use of a high-throughput and high-accuracy phenotyping platform (MARViN grain analyser) was key to uncovering these grain size variations. While routine for major crops such as wheat, this is the first application of high-resolution, image-based grain measurements to a tef panel to our knowledge. This exemplifies the benefits that adopting robust and well-tested phenotyping methodologies from mainstream crops can bring to the research of underutilised crops[10].

Another route to achieving large-grained white tef varieties could be to knock out key transcription factors or enzymes linked with pigmentation in a brown-grained background (through mutation breeding, transgenesis, or genome editing). To implement such an approach, the tef research community will need to increase its understanding of relevant genes and their pleiotropic effects. Contributing to this, we identified a candidate pair of homoeologues present within each of the two major loci for grain size, colour, and the fatty acid EPOD. Et_4A_032842 and Et_4B_039404 are orthologous to *TRANSPARENT TESTA 2* (*TT2*), an R2R3 MYB transcription factor known to regulate seed coat colour in the Brassicaceae family through proanthocyanidin formation and accumulation[39,40].

We propose that variations in the two tef orthologues of *TT2* contribute to variation in grain colour. This aligns with a previous report of two "duplicate" genetic factors *B/b* and *B2/b2* as the major determinants of brown/white pigmentation[57]. Our hypothesis is further supported by our discovery of independent insertions of related LTR-Copia retrotransposons in both homoeologues of the white-grained tef variety Tsedey. These insertions, which are absent in the genome of the brown-grained variety Dabbi, would both lead to truncated and likely non-functional proteins. These *TT2* polymorphisms could also underlie the variation in fatty acid content and grain size, as has been shown in *Arabidopsis* and *Brassica napus*[58,59].

This hypothesis could be tested by mutagenising *TT2* in a brown-grained variety. However, currently, we cannot preclude the association of these traits with alternative or additional candidate genes. The homoeologues Et_4A_032844 and Et_4B_037039 lie within the same two loci and are orthologous to *NF-YA3*, a gene which activates oil accumulation in oil palm mesocarp and increases oil content and seed size when overexpressed in *Arabidopsis*[38]. It is, therefore, possible that the correlated traits (grain colour, size and fatty acid) are modulated by two separate gene families in linkage blocks.

While the above manipulation is an intriguing possibility, we acknowledge that the original cultural preference for white-grained tef varieties is likely linked to flavour and baking properties in addition to aesthetics, although this has not been well-studied. It is therefore also important to study the tef metabolome beyond its direct contribution to colour. Future studies could utilise our extensive metabolome profiling data to conduct a more comprehensive metabolite GWAS (mGWAS) and deepen our understanding of the genetics underpinning differential metabolite accumulation in brown and white-grained tef.

Our work demonstrates the value that can be brought to underutilised crops such as tef by applying resequencing and population genomics in combination with large-scale phenotyping and metabolome profiling. We identify multiple genetic loci for morphological and nutritional traits and suggest how these could inform future research and breeding efforts, including contribution to new tef varieties with desirable plant architecture and consumer-preferred grain traits. Other underutilised crops could also greatly benefit from such methods to accelerate their domestication or improvement.

## Methods

### Germplasm
A core panel of 225 tef accessions was selected from the broader EIAR collection to capture a broad range of phenotypic diversity. This panel consisted of 220 smallholder varieties and 5 improved, registered varieties (Supplementary Data 6). The EIAR collection is originally derived from 2175 tef germplasm accessions[60], 35 cultivars[61], and 10 released improved varieties[62].

### Field phenotyping
The accessions were grown in Ethiopia at three EIAR research sites: Alem Tena, Chefe Donsa, and Debre Zeit. Chefe Donsa and Debre Zeit sites represent non-stressed environments, while Alem Tena represents a moisture-stressed environment (Supplementary Table 6). Each field trial was set up using an augmented block design (Supplementary Data 7) and consisted of 1 m rows sown with 0.3 g grain and spaced 50 cm apart. Most accessions were sown once per field site, but five improved varieties (Ebba, Boset, Bora, Dagim and Felagot) were sown four times per field site as spatial controls.

Data were collected on a range of qualitative and quantitative traits (Supplementary Data 8). Phenotyping methodology was derived from the Tef Breeding Manual[43] and is described in detail in Supplementary Table 7. Qualitative traits included basal stalk colour, grain colour, panicle colour, and panicle form. Quantitative traits included phenology (days to heading, days to maturity, and grain filling period) and agro-morphology (plant height, panicle length, peduncle length, culm length, first culm internode length, second culm internode length, above-ground shoot biomass, grain yield, harvest index, and lodging index). Lodging index was only assessed at Alem Tena and Debre Zeit.

## Grain size measurement
Grain samples from each of the 720 rows across the three field trials were analysed using a MARViN Grain Analyzer. For each sample, 0.075–0.085 g of grain (mean of 262 grains) was evenly distributed on the imaging tray. Mean grain area and TGW were recorded, as well as mean, minimum, and maximum values for grain width and length. Insufficient grain was harvested from accession Trotteriana-T-138 for MARViN analysis.

## DNA extraction and resequencing
For each accession, ~0.7 g fresh leaf tissue was collected from 3-week-old plants from the Alem Tena field trial. DNA was extracted using DNeasy Plant or DNeasy Plant Pro kits (QIAGEN) and eluted in 50 μL AE buffer. Sufficient high-quality DNA could not be extracted for the accession Trotteriana-T-138. For three further accessions (DZ-01-170, DZ-01-1015, Gealamie-T-111), the observed grain colour at Alem Tena did not match that observed at the other two field locations, suggesting heterogeneity in the seed stock. Given that the DNA sample would, therefore, not represent the majority of the phenotyping data, these accessions were not sequenced and therefore not used for GWAS. The remaining 221 DNA samples were sequenced by Novogene UK (Illumina paired-end 150 bp), and data were returned for 220 accessions (no data were produced for DZ-01-12).

## SNP calling, LD calculation, and phylogenetic analyses
Raw sequencing reads were trimmed using fastp[63] and mapped to the *Eragrostis tef* reference genome (cv. Dabbi) using Bowtie2 (v2.4.1)[11,64]. The mapped reads were filtered for MAPQ scores >30 using SAMtools (v1.18), and a VCF file was generated using BCFtools (v1.18)[65–67]. VCF statistics were generated using VCFtools and examined using base R (v4.1.3)[68,69]. Empirically derived filters were applied using VCFtools (--max-missing 0.90; --minQ 30; --minGQ 15; --min-meanDP 10; --max-meanDP 18; --minDP 5; --maxDP 23; --maf 0.025)[70] and linkage pruning was conducted using Plink (v1.90b4.6; --allow-extra-chr; --indep-pairwise 20 5 0.5). The final VCF file of 41,289 SNPs was converted to PHYLIP format using the tool vcf2phylip (v2.9)[71].

TASSEL (v5.2.54)[72] was used to compute pairwise intra-chromosome LD correlation coefficient ($r^2$) between SNP markers across the entire tef genome. LD decay scatterplot was then produced by plotting the $r^2$ values against physical distance (bp) using R software. The intersection point between the genome-wide LD curve and the $r^2$ threshold (0.2) determined the genome-wide LD decay value.

A phylogram was generated using IQ-Tree 2 and arbitrarily rooted against Ada-T-58 based on alphabetical order (v2.3.2; -B 10000; --msub nuclear; -m MFP + ASC; --seed 42)[26]. The phylogram was visualised in R using ggtree (v3.10.1) and ggtreeExtra (v1.12.0)[73,74]. Population structure was investigated by applying ADMIXTURE analysis (v1.3.0, K = 1:20; --cv = 10)[24] and principal component analysis (SNPRelate v1.29.0)[75] to the same VCF file. The results were visualised using Pophelper (v2.3.1) and Plotly (v4.10.1), respectively[76,77]. After defining the redundancy groups, the name of a single accession from each group was arbitrarily assigned to represent the group in subsequent analyses (Supplementary Table 1).

A *k*-mer presence/absence matrix was computed for all 220 sequenced accessions from trimmed reads using scripts from a previously published *k*-mer-GWAS pipeline (https://github.com/wheatgenetics/owwc/tree/master/kGWAS, section 1)[27]. Additional guidance on the use of this

pipeline is provided at https://github.com/quirozcj/kmerGWAS_descriptions. Default parameters were used for all steps. Notably, the default *k*-mer length and minimum *k*-mer frequency per accession were not modified (-m 51 and -L 4, respectively). Shared *k*-mer state rates were computed using custom bash and R scripts (https://github.com/Uauy-Lab/tef_kGWAS_2024).

## Minimal SNP panel selection
The trimmed reads from accessions belonging to redundancy groups were pooled and subsampled down to the average read number per single library (29,350,529 paired-end reads). A VCF file was generated as above for the 150 redundancy groups and singlets. The same filters and linkage pruning were applied. This VCF file was input to the Minimal Marker pipeline[78]. This involves conversion to a genotype matrix, then selection of SNPs. However, prior to SNP selection, the pipeline was modified to convert all heterozygous calls to missing data. Heterozygous loci are unstable between generations and would therefore be unreliable markers for consistently identifying accessions across generations. In contrast, the original target species for the pipeline, apple (*Malus domestica*), is largely propagated vegetatively, so heterozygous loci are stably inherited between generations. Missing calls are not used by the pipeline to distinguish accessions, so this change forced the pipeline to select a panel of SNPs that uniquely identifies the core collection using only loci homozygous across the 150 tef redundancy groups and singlets. The first run selected 14 SNPs fulfilling this remit. To provide redundancy, these SNPs were removed from the genotype matrix and a second run was conducted, leading to the selection of a further 14 SNPs.

An additional consideration was whether the genotypes of the individual members of the redundancy groups were consistent with the overall group genotype. This was investigated using custom Bash and R scripts (https://github.com/Uauy-Lab/tef_kGWAS_2024), and the results are summarised in Supplementary Data 2 and Supplementary Note 1. There were no cases where group members' genotypes compromised the utility of the SNP set for unique identification of the 150 non-redundant accessions.

## Metabolite extraction and profiling
Methanolic metabolite extractions were conducted at the International Livestock Research Institute (ILRI, Ethiopia) on 40 mg grain from each trial plot following a previously published protocol[79]. Briefly, tissue was ground (Tissue Lyser (QIAGEN), 25 Hz, 2 min), added to 1 mL pre-cooled 100% methanol (−20 °C), and placed on ice for 30 min with vortexing every 5 min. The extracts were then centrifuged, vacuum concentrated, and shipped to Aberystwyth University, Wales, UK, for high-resolution metabolite profiling. The samples were resuspended in 300 μL of pre-cooled 100% methanol, vortexed for 5 min and centrifuged at 1000×*g* at 4 °C for 5 min. An aliquot of 200 μL of each sample was used for untargeted metabolite fingerprinting using Flow Infusion Electrospray High-resolution Mass Spectrometry (FIE-HRMS) mode using Q Exactive hybrid quadrupole-Orbitrap mass spectrometer (Thermo-Scientific, UK) where data was captured in negative and positive ionisation mode using Exactive hybrid quadrupole-Orbitrap mass spectrometer (Thermo-Scientific, UK). Quality controls (QC) were derived from a master mix sample where 10 mL of each extract was pooled and also "blanks" of 100% methanol. Three 20 μL injections were performed for each sample as technical replicates. FIE-HRMS metabolite fingerprints in both positive and negative ionisation modes in a single run. 20 μL of samples were injected into a flow of 100 mL min$^{-1}$. The acquisition of mass-to-charge ratio (*m/z*) data and their binning to discrete bins and peaks was conducted as previously described[80,81].

## Statistics and reproducibility
**Statistical analysis on annotated metabolites.** Good-quality metabolite data could not be produced for 15 samples (Supplementary Data 8). A further set of ten samples (including the three Alem Tena discrepancies mentioned previously) was removed because their grain colour at one location did not match the grain colour at the other two locations

(Supplementary Data 8). *m/z* feature intensities from biologically independent samples (brown *n* = 303 and white *n* = 389) were log10 transformed and Pareto scaled, and those differing significantly between brown and white-grained varieties were selected (Student's *t*-test, FDR < 0.05, mass tolerance of 5 ppm). Metabolite identities were assigned to the differential *m/z* features using the Mummichog algorithm[82], with reference to the latest KEGG version of the *Oryza sativa japonica* (RefSeq) metabolite library[28,83]. A mass tolerance of 5 ppm was used, and all possible adducts and isotopes were considered. Where *m/z* values could be matched to multiple metabolites, the metabolite with the smallest mass difference to the *m/z* value was selected. Statistical analysis, principal component analysis (PCA), partial least squares discriminant analysis (PLS-DA), variable metabolite prediction, and volcano plot were carried out using the online R-based platform MetaboAnalyst 6.0 (https://www.metaboanalyst.ca)[84].

**Statistical modelling for BLUP and heritability calculation**. The original field trial design was updated to reflect the treatment of redundant accessions as combined redundancy groups (Supplementary Data 7). Individual plots belonging to the same redundancy group were treated as biological replicates. Data points were removed for the metabolite analyses above. Genotypic BLUPs were calculated using the R package lme4 (v1.1.32)[85] by fitting the following linear mixed model using restricted maximum likelihood (REML):

$$f(Y) = \alpha + \beta X + \gamma Z + \delta W + e$$

Where *Y* is the observed trait value, *f()* is a transformation conducted for normalisation (either square root, natural log, or none), $\alpha$ is the global mean, $\beta$ is location, *X* is a matrix of location effects, $\gamma$ is block by location identity, *Z* is a matrix of block by location effects, $\delta$ is genotype identity, *W* is a matrix of genotype effects, and *e* is the residual. Location was modelled as a fixed effect due to the low number of factor levels, while block by location and genotype were modelled as random effects. The transformation *f()* applied to each trait was selected to make residuals approximately normally distributed and independent of fitted values. Supplementary Tables 2 and 5 describe the transformation applied to each trait and list any additional data points removed for specific traits prior to BLUP calculation. For example, Alem Tena data was removed prior to the calculation of BLUPs for DTH, DTM, and GFP, as this data made the traits unsuitable for linear mixed modelling even after transformation.

Modelling of BLUPs was not deemed appropriate for panicle morphology and grain colour, given their ordinal and binary encoding (respectively) and minimal variation between field sites. Instead, simple means were used as the genotypic values. To rationalise trait values for presentation, the global intercepts were added to the BLUPs for plant height, grain area, and thousand grain weight in Fig. 4a, b. Raw BLUPs were used for kGWAS and SNP-based GWAS computations.

Broad-sense heritability (H²) was calculated via the "Cullis" method[86] (quoted in the Results) and the BLUP-BLUE regression method (i.e., "Walsh and Lynch" method)[87]. Calculations were performed in GenStat[88] using the same transformations as for BLUP derivation. The selected methods are considered robust to unbalanced trial designs and produced similar results (mean difference 0.021, largest difference 0.076). Tef is highly selfing and we have demonstrated very low heterozygosity for this population (mean = 1.5%). Because of this, additive genetic variance ($V_A$) will predominate over dominance variance ($V_D$), so H² is expected to be approximately equal to (though slightly larger than) narrow-sense heritability (h²).

**k-mer-based GWAS**. A new *k*-mer presence/absence matrix was generated as above using the previously pooled and subsampled reads for the 150 accessions or redundancy groups. Association mapping, calculation of significance threshold, and plotting were conducted using the same kGWAS pipeline on 141 accessions (nine redundancy groups were

excluded because they contained both brown and white-grained accessions; Supplementary Table 1)[27]. For the metabolite traits, a further three accessions were excluded because only one datapoint remained for BLUP calculation (DZ-01-517, DZ-01-1376, Hotolla-T-135). *k*-mers were projected onto the Dabbi reference genome and reported as the number of *k*-mers at a given association level per 10 kb genomic bin. The significance threshold for associations was calculated via Bonferroni correction as follows:

$$padj = \frac{0.05}{n/k} = 1.36 \times 10^{-8}$$

Where *n* is the number of *k*-mers utilised for association calculations (187,226,135) and *k* is the *k*-mer length (51). This threshold is plotted as $-\log_{10}(1.36 \times 10^{-8}) = 7.87$ on all Manhattan plots presented. For each trait, putative trait-associated regions were extended from the first bin on a chromosome containing significant *k*-mers and terminated at the point where the subsequent 500 kb contained zero significant *k*-mers. Additional putative regions were then iteratively initiated from the next bin containing significant *k*-mers. Putative regions were then defined as significantly trait-associated if they contained ≥750 significant *k*-mers. Supplementary Table 3 lists all significantly trait-associated regions. Dotplot sequence alignments of the LTR Copia insertions in the candidate genes in these regions were made with the dotplot function in R package SeqinR (v4.2-36)[89].

**SNP-based GWAS**. SNP-based GWAS was carried out via GAPIT (v3)[90] with six different models: FarmCPU, BLINK, MLMM, SUPER, CMLM, and ECMLM. The significance threshold for associations was calculated via Bonferroni correction as follows:

$$padj = \frac{0.05}{n} = 1.01 \times 10^{-6}$$

Where n is the number of SNPs utilised for association calculations (49,660). The VCF file used was the same as that used to generate the minimal SNP panel, except that during linkage pruning, $R^2$ was set to 0.7 instead of 0.5. SNP-trait associations were considered significant when supported by at least two of the six models tested. We also considered nearby SNPs significant if they were supported by different models and separated by less than the LD decay distance (46 kb). Details of all significantly trait-associated SNPs are provided in Supplementary Table 4.

**TT2 Sequence analysis in white and brown-grained tef accessions**
The homoeologous candidate genes, Et_4A_032842 and Et_4B_039404, located in the associated peaks on chr 4A and chr 4B, respectively, were identified as orthologue of *Arabidopsis TT2* gene based on sequence homology. We compared the sequences of tef *TT2* genes between the red-grained accession Dabbi and the white-grained accession, Tsedey. For this, *TT2* genomic sequences (Et_4A_032842 and Et_4B_039404) from the Dabbi reference genome[11] were obtained from Ensembl Plant and were used as query for a BLAST search against the draft genome assembly of Tsedey[12] available at CoGe (https://genomevolution.org/coge/). Scaffolds showing more than 90% percentage identity for each query were extracted from the Tsedey genome assembly using SAMtools faidx tool[67]. *TT2* sequences in the scaffolds were annotated using the gene model for Et_4A_032842 and Et_4B_039404 from the Dabbi assembly (Supplementary Data 4 and 5). To ascertain if the insertions identified within the annotated gene models were RTs, the insertion sequences were used as queries for BLAST search against a database of repeat elements from tef, available at PlantRep[91]. Insertions with more than 90% identity to repeat elements in the database were considered as RTs. The identified RT insertions were manually annotated to highlight the long-terminal repeat and tandem site duplications at either end of the insertions (Supplementary Data 4 and 5).

### Ethics and inclusion statement

This research is highly relevant to local partners in Ethiopia at the Ethiopian Institute for Agricultural Research (EIAR) and the International Livestock Research Institute (ILRI). Both institutes conduct research on and/or breeding of tef, which is a major cereal crop in Ethiopia. Local partners were involved throughout the study design, implementation, data analysis and writing. Roles and responsibilities were agreed upon amongst collaborators during initial grant acquisition processes, though additional researchers became involved throughout the research. Raw data has been made available to all collaborators throughout the research. The field trials for this study were conducted at an existing breeding station in Ethiopia and presented no known additional risks to research staff or the environment. These trials did not require special permissions from local authorities or review by a local ethics review committee. No human or animal research was conducted. Tef DNA and metabolite samples were transferred from Ethiopia to the UK with the Ethiopian government's permission. Raw data obtained from these samples has been shared with local researchers. Regional research has been utilised extensively in this study and cited accordingly.

### Reporting summary

Further information on research design is available in the Nature Portfolio Reporting Summary linked to this article.

### Data availability

Sequencing data is available via NCBI SRA under BioProject ID PRJNA1150514. Raw phenotypic data can be found in the Supplementary Data. Raw metabolomic data, VCF files and source data for graphs and charts in the main figure are available on Zenodo: https://doi.org/10.5281/zenodo.1383731892[92].The tef germplasm used in this study is available for research purposes only upon permission by the Ethiopian government (particularly the Ethiopian Biodiversity Institute) and signing of a Material Transfer Agreement.

### Code availability

Custom Bash and R scripts are available at https://github.com/Uauy-Lab/tef_kGWAS_2024.

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

## Acknowledgements
The authors would like to thank Phil Robinson for help with photography, Martin Vickers for help with depositing sequence data with SRA, and Stephanie Williams for figure design advice. The authors would also like to thank Doni Hinsene and Tadelech Bizuneh for help with greenhouse tef planting and DNA extraction. Metabolite profiling was supported by Manfred Beckmann and Helen Phillips (Aberystwyth University). This work was supported by the Royal Society FLAIR Collaboration Grants 2020 (FCG \R1\201032); Biotechnology and Biological Sciences Research Council (BBSRC) through the Delivering Sustainable Wheat (BB/X011003/1) and Building Robustness in Crops (BB/X01102X/1) Institute Strategic Programmes; European Research Council (ERC-2019-COG-866328); Strategic Program for Resilient Crops: Grains for Health BBSRC grant, BBS/E/IB/230001B; Advancing Plant Health (BB/X010996/1); CGIAR Initiative on Sustainable Animal Productivity for Livelihoods, Nutrition and Gender inclusion (SAPLING). CGIAR research is also supported by contributions to the CGIAR Trust Fund. O.S. was also supported by the Royal Society FLAIR Fellowship (FLR_R1_191850). A.G. was also supported by the European Union's Horizon 2020 research and innovation programme under the Marie Skłodowska-Curie H2020-MSCA-IF-2018 grant agreement No 842118, SUPERTEFF. J.Q.C. was supported by the Mexican Consejo Nacional de Ciencia y Tecnología (CONACYT; 2018-000009-01EXTF-00306) and the JIC International Scholarship (2018–2022).

## Author contributions
M.R.W.J.: Formal Analysis, Software, Investigation, Data Curation, Writing—Original Draft, Visualisation, Project administration; Ab.T.: Formal Analysis, Investigation, Supervision, Writing—Original Draft, Visualisation; A.G.:Formal Analysis, Investigation, Data Curation, Writing—Original Draft, Visualisation; W.K.: Investigation, Data Curation, Resources, Writing—Original Draft; Ad.T.: Investigation, Data Curation, Writing—Original Draft; D.G.: Conceptualisation, Supervision, Resources, Writing—Review & Editing; J.K.M.B.: Formal Analysis, Writing—Review & Editing; J.Q.C.: Methodology, Software; C.S.J.: Resources, Supervision, Writing—Review & Editing; B.B.H.W.: Conceptualisation, Writing—Review & Editing, Funding Acquisition; S.C.: Conceptualisation, Resources, Investigation, Project Administration, Writing—Review & Editing; K.A.: Conceptualisation, Resources, Investigation, Writing—Review & Editing; Z.T.: Conceptualisation, Resources, Writing—Review & Editing; L.A.J.M.: Conceptualisation, Resources, Formal Analysis, Supervision, Writing—Original Draft, Funding Acquisition. C.U.: Conceptualisation, Formal Analysis, Resources, Supervision, Writing—Original Draft, Project Administration, Funding Acquisition; O.S.: Conceptualisation, Formal Analysis, Investigation, Supervision, Writing—Original Draft, Visualisation, Project Administration, Funding Acquisition.

## Competing interests
The authors declare no competing interests.
