## [Transparent Peer Review file · Communications Biology]

Population genomics uncover loci for trait improvement in the indigenous African cereal tef (*Eragrostis tef*)

Corresponding Author: Dr Oluwaseyi Shorinola

Version 0:

Reviewer comments:

Reviewer #1

(Remarks to the Author)

The authors sequenced a Tef population of over 200 accessions and performed GWAS for agronomic and metabolic traits. As a minor crop with significant importance for African people, these genomic resources hold great promise for the improvement and selection of Tef varieties using marker- and genome-assisted breeding. I have the following questions which should be addressed to improve the quality and readability of this work.

1. The authors performed GWAS to identify significant regions associated with agronomic and metabolic traits. Please assess the linkage disequilibrium (LD) of this population to infer the mapping resolution.
2. The authors merged the GWAS results of panicle traits, grain traits and metabolic traits into one section. I suggest to separate them into two or three sections (agronomic & metabolic traits, or panicle, grain and metabolic traits) to improve the readability of these results.
3. It seemed that only a minority of GWAS results by k-mer and SNP marker were overlapped (5 regions, line 245). Please make more analysis to explain this results, for example, the association signals of k-mers that fell within significant loci identified only by SNP-based GWAS. Since the k-mer based GWAS was similar to the SV-based GWAS, was the LD between k-mer and SNPs usually weak?

Minor:

1. The font size of some words in Figure 5 was too small.
2. Line 276-279. "These regions.....These regions". These sentences were weird and should be further revised.

Reviewer #3

(Remarks to the Author)

Dear Editor,

I thank you for being invited to review the manuscript "Population genomics uncover loci for trait improvement in the indigenous African cereal tef (*Eragrostis tef*)"

The manuscript is well-written and summarizes well-performed, well-funded, and well-analyzed experiments. The science, quality, and technique are generally sound.

However, there are significant issues that I believe prevent this manuscript from being suitable for publication in its current form.

1. Motivation and Focus:

The manuscript lacks clarity in articulating the broader motivation behind the research. The authors emphasize tef's relevance within the Horn of Africa but need to acknowledge its global potential and the opportunities it presents as a crop in diverse contexts. Additionally, there seems to be an implicit acceptance of the traditional and often unsustainable agricultural practices in this region without critically examining them.

]

2. Missing Literature:

Some important references are omitted, such as the study available below, which could provide valuable context. Furthermore, the manuscript overlooks significant work on root development and its association with lodging—a critical issue affecting tef yields.

<https://academic.oup.com/plphys/article/192/1/222/7032508>

3. Tef as a Forage Crop:

The manuscript does not address tef's potential as a forage crop or the traits required for optimizing its use in this capacity.

4. Reproducibility and Accessibility of Resources:

Reporting genomic mapping and genetic diversity using germplasm unavailable to the global research community significantly limits the reproducibility and utility of this work. It is up to the editor to determine if this practice aligns with the journal's publication guidelines. However, integrating freely available genetic resources would enhance the study's impact and ensure broader accessibility.

Specific issues with the manuscript- by line number are below:

1. Line 56: The claim that tef is "vital" for Ethiopia warrants reconsideration. While it is undoubtedly important, alternatives such as maize are widely cultivated in Ethiopia. Moreover, the authors could utilize this platform to discuss systemic barriers, such as restrictive governmental policies, that impede agricultural innovation and diversification in tef cultivation

2. Line 74: The reliance on semi-dwarf lines as a solution to lodging reflects a wheat-centric approach that may not fully address tef's agronomic needs. The authors should consider discussing alternative strategies, such as mono-culm traits and improved root anchorage. The current approach ignores the importance of tef straw.

3. Line 75: Could the authors please provide a reference for the actual effect of panicle form in field experiments?

4. Line 77: The authors should acknowledge the trade-off between panicle compactness for mechanical harvesting and the purported lodging reduction benefits of a loose panicle, which remain unsubstantiated.

5. Line 89-91: The discussion on sowing density must avoid perpetuating outdated practices. Suggesting Ethiopian farmers continue broadcasting rather than adopting mechanized agriculture conveys a lack of progressiveness. This must be rephrased to emphasize the need and importance of modern and sustainable farming practices.

6. Line 173: The correlation between seed size and weight is trivial and does not contribute substantively unless presented in a broader context.

7. Line 230: Could the heritability of metabolite clusters be quantified to add rigor?

8. Line 239: Given that tighter panicles were under-represented in the panel, how valid should the associated regions be?

9. Lines 259-267: The extensive list of potential genes could be moved to supplementary material unless accompanied by detailed explanations of their significance.

10. Line 270: Please Avoid discussing grain area and width separately, as they are interrelated metrics. Or explain how they are different.

11. Line 320: Avoid sentimental terms like "orphan" and opt for scientifically precise language.

12. Line 322: If the objective is to support global food systems, the focus should shift toward traits that enhance large-scale production rather than emphasizing traditional, localized traits.

13. Line 338- To genuinely facilitate germplasm exchange, the materials used in the study should be accessible to researchers beyond Ethiopia.

14. 340: This claim has not been shown in replicated field experiments and should not be taken as a known fact.

15. 350: Contrary to the claim about panicle form, the branching control, considering the very high number of tillers produced by the lines, should be considered in relation to lodging.

16. 357- 360: The authors should consider changing the wording here and not assume that farmers in Africa will continue to

use old practices; science cannot bow its head to policy since, in doing so, it perpetuates the suffering and lack of development of the very people it should strive to help. If there is a conflict of interest by the researchers in calling for more advanced agricultural practices to be used in a country where food shortages are still happening, it should be described.

17. Line 360: The Authors should clarify why they define the lodging in teff as “stem lodging” or remove this claim previously shown to be questionable. Teff plants that lodge do so while uprooting the roots of one side of the plants- which is defined as Root lodging.

18. Line 361: The claim about surface sowing affecting root depth appears questionable. Reference studies to substantiate or revise this statement. Teff has extremely deep roots. The claim about root lodging affected by surface sowing seems correct but requires a reference.

19. Line 380: Prior literature identifies grain color as controlled by two gene sets. The authors should engage with this work to validate or refute these findings:

Berhe, Tareke. Inheritance of lemma color, seed color and panicle form among four cultivars of *Eragrostis tef* (Zucc.) Trotter. The University of Nebraska-Lincoln, 1981. This also suggests that the decision to use two colors is not well established.

20. Line 394: Specify the criteria for distinguishing between "brown" and "white" grains in general and in image analysis (did you use RGB?)

21. Line 468: Can you provide data on the tiller number, as its absence raises concerns?

22. Line 470: Explain how the lodging index was calculated.

23. Line 500: Was the phylogram explored using additional rooting methods? Alphabetical order appears unconventional.

24. Line 564: Could the authors clarify the distinction between the lines mentioned here and those in line 484, especially given the focus on grain color?

25. Line 630: Please Justify using the multiplication factor 1.01×10^{-6} with an appropriate reference or explanation.

Version 1:

Reviewer comments:

Reviewer #1

(Remarks to the Author)

The authors have made great efforts to address my concerns. I am happy to see the re-organization of GWAS results into 3 different subtitles, and the clarification of LD between k-mers and SNPs. The manuscript is significantly improved and ready to be accepted by Communications Biology.

Reviewer #3

(Remarks to the Author)

I thank the authors for taking the time and investing the effort to address some of the issues this great manuscript had.

While I do not agree with their opinions on all topics, I understand their opinion and accept it.

My concerns have been adequately corrected.

This is an important contribution to the scientific community and the teff research community specifically.

Good luck

16 February 2025,

Dear Communications Biology Editorial Team,

We thank both reviewers for their careful analyses of our work. We aim to address their points in our responses below (green text).

Reviewer #1 (Remarks to the Author):

The authors sequenced a Tef population of over 200 accessions and performed GWAS for agronomic and metabolic traits. As a minor crop with significant importance for African people, these genomic resources hold great promise for the improvement and selection of Tef varieties using marker- and genome-assisted breeding. I have the following questions which should be addressed to improve the quality and readability of this work.

1. The authors performed GWAS to identify significant regions associated with agronomic and metabolic traits. Please assess the linkage **disequilibrium (LD) of this population to infer the mapping resolution.**

Thank you for this suggestion. Using our SNP dataset, we have now performed pairwise LD (R^2) estimation across the genome and used these estimates to create a LD decay plot (Supplementary Figure 1). Based on the plot, we estimate that the average mapping resolution in the population is ~46 kb. However, mapping resolution will vary across the genome. So, we have also defined an approximate mapping resolution for each of our associated loci using a heuristic approach based on the p-value and number of significant k -mers in each successive 10 kb bin around the significant region. The mapping resolution of significant regions is already presented in Supplementary Table 6.

2. The authors merged the GWAS results of panicle traits, grain traits and metabolic traits into one section. I suggest to separate them into two or three sections (agronomic & metabolic traits, or panicle, grain and metabolic traits) to improve the readability of these results.

Thank you for this suggestion. We have now separated the GWAS results section into three sections that describe the GWAS results for agronomic traits, GWAS results for metabolite traits, and candidate gene analyses.

3. It seemed that only a minority of GWAS results by k -mer and SNP marker were overlapped (5 regions, line 245). Please make more analysis to explain this results, for example, the association signals of k -mers that fell within significant loci identified only by SNP-based GWAS. Since the k -mer based GWAS was similar to the SV-based GWAS, was the LD between k -mer and SNPs usually weak?

We thank the reviewer for this suggestion. We have re-analysed the result of our SNP-based GWAS only to select MTAs that are supported by at least two of the six statistical models used for GWAS association: BLINK, SUPER, FarmCPU, ECMLM, CMLM, MLMM. This criterion improves the robustness and

We advance

We activate

confidence in our SNP GWAS. It reduces the number of MTAs from 91 to 13, five (~40%) overlapping with the significant regions identified from our *k*-mer GWAS.

In addition, as the reviewer rightly suggested, some of the significant regions identified from our *k*-mer GWAS likely arise from structural variations that cannot be identified in SNP-based analysis. A perfect example of this in our study is the transposable element insertions in *TT2* that we identified in our *k*-mer GWAS but in the SNP-based GWAS.

Minor:

1. The font size of some words in Figure 5 was too small.

Thank you. We have addressed this point in our revised figures.

2. Line 276-279. "These regions.....These regions". These sentences were weird and should be further revised.

Thank you, we agree. This has been resolved with the restructuring suggested in major point 2.

Reviewer #3 (Remarks to the Author):

Dear Editor,

I thank you for being invited to review the manuscript "Population genomics uncover loci for trait improvement in the indigenous African cereal tef (*Eragrostis tef*)"

The manuscript is well-written and summarizes well-performed, well-funded, and well-analyzed experiments. The science, quality, and technique are generally sound.

However, there are significant issues that I believe prevent this manuscript from being suitable for publication in its current form.

1. Motivation and Focus:

The manuscript lacks clarity in articulating the broader motivation behind the research. The authors emphasize tef's relevance within the Horn of Africa but need to acknowledge its global potential and the opportunities it presents as a crop in diverse contexts. Additionally, there seems to be an implicit acceptance of the traditional and often unsustainable agricultural practices in this region without critically examining them.

We appreciate the reviewer's feedback and acknowledge the need for greater clarity on our motivation. The motivation of our study is to use a population genomics approach to dissect the genetic underpinnings of economically important traits in tef. We have now clearly defined this aim in the introduction. We have also added additional text to describe further the wider adoption of tef beyond Ethiopia and the alternative use of tef as a forage crop.

Our manuscript describes traditional tef cultivation practices (particularly manual broadcasting), recognising that millions of resource-poor smallholder farmers in Ethiopia employ these practices. However, we do not affirm or oppose these practices but rather aim to highlight the challenges faced by these farmers and how the findings from our work can help address these. We have modified our manuscript to clarify our discussion of these points.

We advance

We activate

2. Missing Literature:

Some important references are omitted, such as the study available below, which could provide valuable context. Furthermore, the manuscript overlooks significant work on root development and its association with lodging—a critical issue affecting tef

yields. <https://academic.oup.com/plphys/article/192/1/222/7032508>

We agree that the proposed reference is an exciting study. However, we are unclear of the relevance of this paper to our work. The paper explores and presents a compelling argument that there is a novel mechanism and novel genetics underlying grain shattering in tef and perhaps *Eragrostis* species more widely. However, grain and rachis shattering are not traits we phenotyped and consequently we did not present literature on them. We have, however, taken the reviewer's comments on board by including references to papers exploring root traits for lodging resistance. Given that root traits were beyond the scope of the present study, we did not go into an in-depth discussion on this topic.

3. Tef as a Forage Crop:

The manuscript does not address tef's potential as a forage crop or the traits required for optimizing its use in this capacity.

We acknowledge and have further discussed using tef as a forage crop in our revised manuscript. We would like to highlight that it was impossible to phenotype all traits within the present study. As a consortium, including local Ethiopian partners, we decided to focus this large study on agronomic and morphological traits that affect grain yield and quality. Importantly, the genomics resources we have developed could be used in the future to study other traits that we did not have the resources to phenotype in the present study. This includes traits that improve the forage potential of tef.

4. Reproducibility and Accessibility of Resources:

Reporting genomic mapping and genetic diversity using germplasm unavailable to the global research community significantly limits the reproducibility and utility of this work. It is up to the editor to determine if this practice aligns with the journal's publication guidelines. However, integrating freely available genetic resources would enhance the study's impact and ensure broader accessibility.

We are sorry this was unclear in our initial submission. The tef germplasm used in this study are all available for research purposes by the Ethiopian government through the Ethiopian Biodiversity Institute upon signing of a Material Transfer Agreement (MTA). This is common practice and in line with international guidelines. We have now added this statement to the data availability section of the manuscript.

In addition, the genomic mapping and *k*-mer analyses described in our manuscript are transferable to other tef populations. We are currently in the process of validating some of the candidate loci described in our manuscript in the independent USDA tef germplasm collection. We believe that the findings reported in our manuscript will be relevant and of interest to researchers working on other tef collections globally, as evidenced by the multiple communications we received from colleagues after our pre-print was posted online.

Specific issues with the manuscript- by line number are below:

1. Line 56: The claim that tef is "vital" for Ethiopia warrants reconsideration. While it is undoubtedly important, alternatives such as maize are widely cultivated in Ethiopia. Moreover, the authors could utilize this platform to discuss systemic barriers, such as restrictive governmental policies, that impede

We advance

We activate

We have rephrased this sentence as “tef is an indigenous staple crop in Ethiopia...”. In addition, we have now included a statement in the introduction to highlight government policies on tef export and its impact on the broader adoption of tef. However, we believe that extensively discussing these policy issues is outside the scope of our genetics research. Instead, we have referred readers to reviews and policy papers that describe the impact of these policies.

2. Line 74: The reliance on semi-dwarf lines as a solution to lodging reflects a wheat-centric approach that may not fully address teff's agronomic needs. The authors should consider discussing alternative strategies, such as mono-culm traits and improved root anchorage. The current approach ignores the importance of teff straw.

We agree with the reviewer that there are many different approaches to addressing lodging in tef. In our manuscript, we have not favoured or advocated for using semi-dwarfism to address lodging. Instead, we stated that semi-dwarfism has had limited success in reducing lodging in tef. We believe this aligns with the reviewer's view on not over-depending on semi-dwarf lines. Nonetheless, we have now rephrased this section to highlight other approaches for addressing lodging in tef and to highlight that semi-dwarfism could compromise forage yields. Thank you.

3. Line 75: Could the authors please provide a reference for the actual effect of panicle form in field experiments?

We have provided a reference at the end of the paragraph for a study that shows the impact of panicle form on lodging resistance. The study combined plant phenotyping (lodging, plant height, panicle angle and weight) under controlled conditions, mechanical testing of stem strength and crop modelling to show that panicle angle is an important factor for stem lodging. We, however, recognise that this study was conducted under a controlled environment and might not fully capture the environmental complexity of field-grown plants. In the paragraph, we have now described the experimental context for our assertion.

4. Line 77: The authors should acknowledge the trade-off between panicle compactness for mechanical harvesting and the purported lodging reduction benefits of a loose panicle, which remain unsubstantiated.

We acknowledge that there has not been extensive research on the impact of panicle form on lodging. However, the limited evidence currently available suggests that panicle laxness increases rather than reduces lodging (Blösch et al., 2020, *Frontiers in Plant Science*). Also, while we have no direct evidence for how different panicle forms in tef are amenable to mechanical harvesting, studies in rice suggest that closed panicles (similar to compact panicles) enhance seed retention by reducing seed shattering (Ishikawa et al., 2022, *PNAS*). This implies that a compact panicle ideotype might benefit lodging reduction and mechanical harvesting.

5. Line 89-91: The discussion on sowing density must avoid perpetuating outdated practices. Suggesting Ethiopian farmers continue broadcasting rather than adopting mechanized agriculture conveys a lack of progressiveness. This must be rephrased to emphasize the need and importance of modern and sustainable farming practices.

We recognise current Ethiopian agricultural systems and their ongoing transformation. In our revised discussion, we have emphasized both the existing realities and the dynamic changes shaping the sector.

We advance

We activate

While smallholder farmers managing plots of 0.25 to 1 hectare predominantly rely on traditional methods, the adoption of mechanisation is progressively advancing. Large tef grains aligned with this modernisation effort offer support that can be integrated into both current and emerging agricultural practices.

6. Line 173: The correlation between seed size and weight is trivial and does not contribute substantively unless presented in a broader context.

We believe it is worth briefly mentioning the confirmation of an expected trend by our data. Furthermore, this statement then leads into our identification of a bimodal distribution of grain size versus weight well-explained by grain colour. This is a novel, non-trivial finding in tef and suggests that white-grained varieties produce higher-density grains.

7. Line 230: Could the heritability of metabolite clusters be quantified to add rigor?

We have now quantified the Cullis and BLUP-BLUE broad-sense heritability of each trait analysed by GWAS, including the 21 metabolites. We have discussed the added heritability results in the revised text and included all values in Supplementary Table 4 and 8. We thank the reviewer for this suggestion.

8. Line 239: Given that tighter panicles were under-represented in the panel, how valid should the associated regions be?

The detected association is statistically significant and is, therefore, valid. We refer the reviewer to the insightful discussion by Wang and Xu, (2019, *Heredity*, <https://doi.org/10.1038/s41437-019-0205-3>) that argues that: "if a QTL is statistically significant, it is true (subject to the controlled Type 1 error), regardless of how small the sample size is". We employed a very conservative (Bonferroni) correction method to control for type 1 error. We are, therefore, confident that the detected association is not a type 1 error and is valid. The fact that we still detect a significant region suggests a strong genotype-phenotype association that compensates for the low frequency of the accessions with the tightest panicle category.

Furthermore, it should be noted that we used an ordinal scale for scoring the panicle laxness, scored as 1 - 4 with increasing levels of compactness. The allele for compactness is therefore contributed from different panicle type categories. That is, the significant association detected is for the quantitative difference between all four categories.

9. Lines 259-267: The extensive list of potential genes could be moved to supplementary material unless accompanied by detailed explanations of their significance.

We have not provided an extensive list of potential genes in this section. We describe one potential candidate gene (*CYP931G*) for the region associated with accumulation of kaempferol 3-O-rhamnoside-7-O-glucoside (KOROG). We believe it is reasonable to mention this candidate gene, given its previously reported role in the biosynthesis of flavonol glycosides.

10. Line 270: Please Avoid discussing grain area and width separately, as they are interrelated metrics. Or explain how they are different.

We agree that they are interrelated metrics. However, given that grain area is influenced by both length and width, it is an interesting observation that the detected regions for grain area were more strongly driven by width. Another marginally insignificant region for grain length and grain area was observed on chromosome 10A, with a near total absence of signal for grain width.

We think it is worth discussing both grain width and area in relation to the detected 4A and 4B peaks, as there is no guarantee that a significant grain width or length effect is large enough to translate into a significant grain area effect. We prefer to present both effects to allow readers the full information required for their own interpretation.

We advance

We activate

11. Line 320: Avoid sentimental terms like "orphan" and opt for scientifically precise language.

We utilised the term in reference to the recent Nature Portfolio special issue call "Orphan crop genomics and improvement". We have now referred to these as underutilised.

12. Line 322: If the objective is to support global food systems, the focus should shift toward traits that enhance large-scale production rather than emphasizing traditional, localized traits.

Given that currently, more than 90% of global tef production is in Ethiopia, our trait selection strategy was informed by extensive consultation and collaboration with local researchers in Ethiopia. This ensures that our work will have maximum impact on tef cultivation where it is most produced.

In addition, we argue that many of the traits examined are very relevant for tef cultivation everywhere (inside and outside of Ethiopia). We measured 17 agronomic traits ranging from phenology traits (flowering time and maturity time), morphological traits (panicle form and colour, and plant height), and yield-related traits (biomass, grain yield and grain size). In addition, recognising the importance of addressing nutritional security, we measured the concentration of grain metabolites. We analysed 21 metabolites in greater detail, many of which are known to be involved in grain pigmentation or important for human nutrition.

Many of these traits are primary improvement targets for breeders globally in many different crops, not just tef. While we found significant marker-trait association for only a few of these traits, this should not be taken to imply that we only attempted to study a few traits. Furthermore, our work shows a relationship between grain size, grain colour, and grain metabolite content not previously shown in tef. These relationships are critical for balancing the need for large-scale production (increased yield) and enhancing nutrient content (grain quality), thus ensuring both food and nutrition security for global food systems.

13. Line 338- To genuinely facilitate germplasm exchange, the materials used in the study should be accessible to researchers beyond Ethiopia.

As we describe in our earlier response to major point 4, the tef germplasm used in this study are available for research purposes by the Ethiopian government via the Ethiopian Biodiversity Institute upon signing an MTA.

14. 340: This claim has not been shown in replicated field experiments and should not be taken as a known fact.

We have already provided a reference and described the experimental context for our claim on the effect of panicle angle on lodging in the introduction. We have however taken the reviewer's comment on board and have removed the sentence so as to focus our discussion on the genetic control of panicle form.

15. 350: Contrary to the claim about panicle form, the branching control, considering the very high number of tillers produced by the lines, should be considered in relation to lodging.

We agree with the reviewer that tiller number is an important trait to consider in tef. However, given the scale of our field phenotyping (220 accessions x 17 agronomic traits x 3 locations, plus metabolite sampling) and the limited resources available to us, we were not able to measure every relevant trait in tef. This includes tiller number, which is a labour-intensive trait to measure. The genomic data we have generated in this study will be useful in dissecting the regulation of tiller numbers should phenotypic data for this become available in the future.

16. 357- 360: The authors should consider changing the wording here and not assume that farmers in Africa will continue to use old practices; science cannot bow its head to policy since, in doing so, it perpetuates the suffering and lack of development of the very people it should strive to help. If there is a conflict of interest by the researchers in calling for more advanced agricultural practices to be used in a

We advance

We activate

country where food shortages are still happening, it should be described.

We make no comments, positive or negative, on the relative merits of different approaches to farming as we do not believe that is our role when producing original biological research. The reviewers' comment relates to policy and social science and may be more suitable in response to a review article. We maintain that we have no conflicts of interest relating to our research.

17. Line 360: The Authors should clarify why they define the lodging in teff as "stem lodging" or remove this claim previously shown to be questionable. Teff plants that lodge do so while uprooting the roots of one side of the plants- which is defined as Root lodging.

Several authors on this paper have experienced both stem and root lodging in tef, hence our inclusion of both categories in our discussions. Nonetheless, we did not quantify these modalities separately in this work and, in the absence of robust published data on stem lodging, we agree that we should remove this.

18. Line 361: The claim about surface sowing affecting root depth appears questionable. Reference studies to substantiate or revise this statement. Teff has extremely deep roots. The claim about root lodging affected by surface sowing seems correct but requires a reference.

We have now revised this paragraph and removed this statement.

19. Line 380: Prior literature identifies grain color as controlled by two gene sets. The authors should engage with this work to validate or refute these findings:

Berhe, Tareke. Inheritance of lemma color, seed color and panicle form among four cultivars of *Eragrostis tef* (Zucc.) Trotter. The University of Nebraska-Lincoln, 1981. This also suggests that the decision to use two colors is not well established.

The literature identified by the reviewer is consistent with our findings of two major candidate regions at homoeologous loci for grain colour. We thank them for bringing this work to our attention and have engaged with this research in our updated Discussion.

Regarding the latter point, we also categorise grain colour ordinally into five categories: dark brown, brown, pale white, white, and very white. The GWAS results deriving from this ordinal classification were highly similar to those using a binary classification. However, for the final analyses presented, we felt that the sub-division of colour into more than two categories would be unreliable and too susceptible to environmental variation. The paper cited by the reviewer itself acknowledges "difficulties met in classifying dark and medium browns".

20. Line 394: Specify the criteria for distinguishing between "brown" and "white" grains in general and in image analysis (did you use RGB?).

Brown and white grains were classified by eye by two researchers with extensive experience working with tef. We feel this approach is suitably robust for binary grain colour classification but agree that image analysis would be beneficial for further subdivision.

21. Line 468: Can you provide data on the tiller number, as its absence raises concerns?

See response to comment 15.

22. Line 470: Explain how the lodging index was calculated.

A full description of all phenotyping methodologies is included in Supplementary Table 13. Lodging index was calculated according to Caldicott and Nuttall (1979):

$$\text{Lodging index} = \sum ((\text{lodging degree} * \text{percent of plot affected}) / 5)$$

We advance

We activate

Where 'lodging degree' varies from 0 for completely upright to 5 for completely lodged (flat against the ground). The calculated values for lodging index therefore vary between 0 (no lodging) and 100 (complete lodging).

23. Line 500: Was the phylogram explored using additional rooting methods? Alphabetical order appears unconventional.

We attempted to root the phylogram against an outgroup, *Eragrostis curvula*, using short-read sequencing data from Carballo et al., 2023 *Front Plant Sci.* (NCBI SRA SRR22846089). Unfortunately, after aligning this data to the *E. tef* 'Dabbi' reference genome, insufficient SNPs could be called to enable inclusion in the phylogram.

Rooting against the alphabetically foremost accession is indeed arbitrary, but it is the default in the absence of a known root for the popular package we employed (IQTree 2). We also generated an unrooted phylogram for comparison but found that it was less compact and less readily annotatable. It, therefore, ultimately failed to convey the main message intended for the figure: that there was considerable redundancy within the studied germplasm collection.

Given this aim for the figure - as opposed to a detailed exploration of the evolutionary and domestication history of this collection - we feel we are justified in employing an arbitrary root for the tree, as the accurate reconstruction of major clades was not our goal. We include a note in the figure legend to highlight the use of this arbitrary root.

24. Line 564: Could the authors clarify the distinction between the lines mentioned here and those in line 484, especially given the focus on grain color?

Apologies, we agree that this was unclear and have now added a clarification to line 564. In total, there were 10 accessions which displayed a different grain colour at one location versus the other two. These 10 individual accession-location data points were removed from metabolite analyses and GWAS.

Of these 10 accessions, three displayed a different grain colour at Alem Tena versus the two other sites. Given that DNA was collected from Alem Tena plants only, the sequencing data we generated would not have corresponded to the majority of the phenotyping data. We therefore decided not to sequence these three lines and so they could not be included in our GWAS analyses.

25. Line 630: Please Justify using the multiplication factor 1.01×10^{-6} with an appropriate reference or explanation.

This number is not a multiplication factor; it is the final significance threshold calculated for SNP GWAS. The equation describes its calculation through the division of an original p-value threshold (0.05) by the number of SNPs utilised (49,660) - i.e., application of a Bonferroni correction.

Yours faithfully,

Oluwaseyi Shorinola
(On behalf of Jones et al.)

Dear Editor,

I thank you for being invited to review the manuscript “*Population genomics uncover loci for trait improvement in the indigenous African cereal tef (Eragrostis tef)*”

The manuscript is well-written and summarizes well-performed, well-funded, and well-analyzed experiments. The science, quality, and technique are generally sound.

However, there are significant issues that I believe prevent this manuscript from being suitable for publication in its current form.

1. Motivation and Focus:

The manuscript lacks clarity in articulating the broader motivation behind the research. The authors emphasize tef's relevance within the Horn of Africa but need to acknowledge its global potential and the opportunities it presents as a crop in diverse contexts. Additionally, there seems to be an implicit acceptance of the traditional and often unsustainable agricultural practices in this region without critically examining them.

]

2. Missing Literature:

Some important references are omitted, such as the study available below, which could provide valuable context. Furthermore, the manuscript overlooks significant work on root development and its association with lodging—a critical issue affecting tef yields.

<https://academic.oup.com/plphys/article/192/1/222/7032508>

3. Tef as a Forage Crop:

The manuscript does not address tef's potential as a forage crop or the traits required for optimizing its use in this capacity.

4. Reproducibility and Accessibility of Resources:

Reporting genomic mapping and genetic diversity using germplasm unavailable to the global research community significantly limits the reproducibility and utility of this work. It is up to the editor to determine if this practice aligns with the journal's publication guidelines. However, integrating freely available genetic resources would enhance the study's impact and ensure broader accessibility.

Specific issues with the manuscript- by line number are below:

1. Line 56: The claim that tef is "vital" for Ethiopia warrants reconsideration. While it is undoubtedly important, alternatives such as maize are widely cultivated in Ethiopia. Moreover, the authors could utilize this platform to discuss systemic barriers, such as restrictive governmental policies, that impede agricultural innovation and diversification in tef cultivation

2. Line 74: The reliance on semi-dwarf lines as a solution to lodging reflects a wheat-centric approach that may not fully address teff's agronomic needs. The authors should consider discussing alternative strategies, such as mono-culm traits and improved root anchorage. The current approach ignores the importance of teff straw.
3. Line 75: Could the authors please provide a reference for the actual effect of panicle form in field experiments?
4. Line 77: The authors should acknowledge the trade-off between panicle compactness for mechanical harvesting and the purported lodging reduction benefits of a loose panicle, which remain unsubstantiated.
5. Line 89-91: The discussion on sowing density must avoid perpetuating outdated practices. Suggesting Ethiopian farmers continue broadcasting rather than adopting mechanized agriculture conveys a lack of progressiveness. This must be rephrased to emphasize the need and importance of modern and sustainable farming practices.
6. Line 173: The correlation between seed size and weight is trivial and does not contribute substantively unless presented in a broader context.
7. Line 230: Could the heritability of metabolite clusters be quantified to add rigor?
8. Line 239: Given that tighter panicles were under-represented in the panel, how valid should the associated regions be?
9. Lines 259-267: The extensive list of potential genes could be moved to supplementary material unless accompanied by detailed explanations of their significance.
10. Line 270: Please Avoid discussing grain area and width separately, as they are interrelated metrics. Or explain how they are different.
11. Line 320: Avoid sentimental terms like "orphan" and opt for scientifically precise language.
12. Line 322: If the objective is to support global food systems, the focus should shift toward traits that enhance large-scale production rather than emphasizing traditional, localized traits.
13. Line 338- To genuinely facilitate germplasm exchange, the materials used in the study should be accessible to researchers beyond Ethiopia.

14. 340: This claim has not been shown in replicated field experiments and should not be taken as a known fact.
15. 350: Contrary to the claim about penicle form, the branching control, considering the very high number of tillers produced by the lines, should be considered in relation to lodging.
16. 357- 360: The authors should consider changing the wording here and not assume that farmers in Africa will continue to use old practices; science cannot bow its head to policy since, in doing so, it perpetuates the suffering and lack of development of the very people it should strive to help. If there is a conflict of interest by the researchers in calling for more advanced agricultural practices to be used in a country where food shortages are still happening, it should be described.
17. Line 360: The Authors should clarify why they define the lodging in teff as “stem lodging” or remove this claim previously shown to be questionable. Teff plants that lodge do so while uprooting the roots of one side of the plants- which is defined as Root lodging.
18. Line 361: The claim about surface sowing affecting root depth appears questionable. Reference studies to substantiate or revise this statement. Teff has extremely deep roots. The claim about root lodging affected by surface sowing seems correct but requires a reference.
19. Line 380: Prior literature identifies grain color as controlled by two gene sets. The authors should engage with this work to validate or refute these findings:

Berhe, Tareke. Inheritance of lemma color, seed color and panicle form among four cultivars of Eragrostis tef (Zucc.) Trotter. The University of Nebraska-Lincoln, 1981. This also suggests that the decision to use two colors is not well established.
20. Line 394: Specify the criteria for distinguishing between "brown" and "white" grains in general and in image analysis (did you use RGB?)
21. Line 468: Can you provide data on the tiller number, as its absence raises concerns?
22. Line 470: Explain how the lodging index was calculated.
23. Line 500: Was the phylogram explored using additional rooting methods? Alphabetical order appears unconventional.
24. Line 564: Could the authors clarify the distinction between the lines mentioned here and those in line 484, especially given the focus on grain color?

25. Line 630: Please Justify using the multiplication factor 1.01×10^{-6} with an appropriate reference or explanation.